# ViViT: Curvature Access Through The Generalized Gauss-Newton's Low-Rank Structure

**Felix Dangel**[*]
*University of Tübingen, Tübingen, Germany*

*f.dangel@uni-tuebingen.de*

**Lukas Tatzel**[*]
*University of Tübingen, Tübingen, Germany*

*lukas.tatzel@uni-tuebingen.de*

**Philipp Hennig**
*University of Tübingen & MPI for Intelligent Systems, Tübingen, Germany*

*philipp.hennig@uni-tuebingen.de*

**Reviewed on OpenReview:** *https://openreview.net/forum?id=DzJ7JfPXkE*

## Abstract

Curvature in form of the Hessian or its generalized Gauss-Newton (GGN) approximation is valuable for algorithms that rely on a local model for the loss to train, compress, or explain deep networks. Existing methods based on implicit multiplication via automatic differentiation or Kronecker-factored block diagonal approximations do not consider noise in the mini-batch. We present ViViT, a curvature model that leverages the GGN's low-rank structure without further approximations. It allows for efficient computation of eigenvalues, eigenvectors, as well as per-sample first- and second-order directional derivatives. The representation is computed in parallel with gradients in one backward pass and offers a fine-grained cost-accuracy trade-off, which allows it to scale. We demonstrate this by conducting performance benchmarks and substantiate ViViT's usefulness by studying the impact of noise on the GGN's structural properties during neural network training.

## 1 Introduction & Motivation

The large number of trainable parameters in deep neural networks imposes computational constraints on the information that can be made available to optimization algorithms. Standard machine learning libraries (Abadi et al., 2015; Paszke et al., 2019) mainly provide access to first-order information in the form of *average* mini-batch gradients. This is a limitation that complicates the development of novel methods that may outperform the state-of-the-art: They must use the same objects to remain easy to implement and use, and to rely on the highly optimized code of those libraries. There is evidence that this has led to stagnation in the performance of first-order optimizers (Schmidt et al., 2021). Here, we thus study how to provide efficient access to richer information, namely higher-order derivatives and their distribution across the mini-batch.

Recent advances in automatic differentiation (Bradbury et al., 2020; Dangel et al., 2020) have made such information more readily accessible through vectorization of algebraic structure in the differentiated loss. We leverage and extend this functionality to efficiently access curvature in form of the Hessian's generalized Gauss-Newton (GGN) approximation. It offers practical advantages over the Hessian and is established for training (Martens, 2010; Martens & Grosse, 2015), compressing (Singh & Alistarh, 2020), or adding uncertainty to (Ritter et al., 2018b;a; Kristiadi et al., 2020) neural networks. It is also linked theoretically to the natural gradient method (Amari, 2000) via the Fisher information matrix (Martens, 2020, Section 9.2).

Traditional ways to access curvature fall into two categories. Firstly, repeated automatic differentiation allows for matrix-free exact multiplication with the Hessian (Pearlmutter, 1994) and GGN (Schraudolph, 2002).

---

[*]Equal contribution

Iterative linear and eigensolvers can leverage such functionality to compute Newton steps (Martens, 2010; Zhang et al., 2017; Gargiani et al., 2020) and spectral properties (Sagun et al., 2017; 2018; Adams et al., 2018; Ghorbani et al., 2019; Papyan, 2019b; Yao et al., 2019; Granziol et al., 2021) on arbitrary architectures thanks to the generality of automatic differentiation. However, repeated matrix-vector products are potentially detrimental to performance.

Secondly, K-FAC (Kronecker-factored approximate curvature) (Martens & Grosse, 2015; Grosse & Martens, 2016; Botev et al., 2017; Martens et al., 2018) constructs an explicit light-weight representation of the GGN based on its algebraic Kronecker structure. The computations are streamlined via gradient backpropagation and the resulting matrices are cheap to store and invert. This allows K-FAC to scale: It has been used successfully with large mini-batches (Osawa et al., 2019). One reason for this efficiency is that K-FAC only approximates the GGN's block diagonal, neglecting interactions across layers. Such terms could be useful, however, for applications like uncertainty quantification with Laplace approximations (Ritter et al., 2018b;a; Kristiadi et al., 2020; Daxberger et al., 2021) that currently rely on K-FAC. Moreover, due to its specific design for optimization, the Kronecker representation does not become more accurate with more data. It remains a simplification, exact only under assumptions unlikely to be met in practice (Martens & Grosse, 2015). This might be a downside for applications that depend on a precise curvature proxy.

Here, we propose ViViT (inspired by $\boldsymbol{V}\boldsymbol{V}^\top$ in Equation (3)), a curvature model that leverages the GGN's low-rank structure. Like K-FAC, its representation is computed in parallel with gradients. But it allows a cost-accuracy trade-off, ranging from the *exact* GGN to an approximation that has the cost of a single gradient computation. Our contributions are as follows:

- We present how to compute various GGN properties efficiently by exploiting its low-rank structure: The exact eigenvalues, eigenvectors, and per-sample directional derivatives (Figure 1). In contrast to other methods, these quantities allow modeling curvature noise.

- We introduce approximations that allow a flexible trade-off between computational cost and accuracy, and provide a fully-featured efficient implementation in PyTorch (Paszke et al., 2019) on top of the BackPACK (Dangel et al., 2020) package at `https://github.com/f-dangel/vivit`.

- We empirically demonstrate scalability and efficiency of leveraging the GGN's low-rank structure through benchmarks on different deep architectures. Finally, we use the previously inaccessible properties of the GGN to study how it is affected by noise during training.

The main focus of this work is demonstrating that many interesting curvature properties, including uncertainty, can be computed efficiently. Practical applications are discussed in Section 5.

## 2 Notation & Method

**Setting:** Consider a model $f : \Theta \times \mathbb{X} \to \mathbb{Y}$ and a dataset $\mathcal{D}$ of tuples $(\boldsymbol{x}, \boldsymbol{y}) \in \mathbb{X} \times \mathbb{Y}$. The network, parameterized by $\boldsymbol{\theta} \in \Theta$, maps a sample $\boldsymbol{x}$ to a prediction $\hat{\boldsymbol{y}} = f(\boldsymbol{\theta}, \boldsymbol{x})$. Predictions are scored by a convex loss function $\ell : \mathbb{Y} \times \mathbb{Y} \to \mathbb{R}$ (e.g. cross-entropy or square loss), which compares to the ground truth $\boldsymbol{y}$. The training objective is the empirical risk $\frac{1}{|\mathcal{D}|} \sum_{(\boldsymbol{x}, \boldsymbol{y}) \in \mathcal{D}} \ell(f(\boldsymbol{\theta}, \boldsymbol{x}), \boldsymbol{y})$. In the following, we consider the loss $\mathcal{L} : \Theta \to \mathbb{R}$

$$\mathcal{L}(\boldsymbol{\theta}) = \tfrac{1}{N} \sum_{n=1}^{N} \ell(f(\boldsymbol{\theta}, \boldsymbol{x}_n), \boldsymbol{y}_n)\,, \tag{1}$$

evaluated on a mini-batch $\{(\boldsymbol{x}_n, \boldsymbol{y}_n) \in \mathbb{X} \times \mathbb{Y}\}_{n=1}^{N} \subset \mathcal{D}$ with $N$ samples. We use $\ell_n(\boldsymbol{\theta}) = \ell(f(\boldsymbol{\theta}, \boldsymbol{x}_n), \boldsymbol{y}_n)$ and $f_n(\boldsymbol{\theta}) = f(\boldsymbol{\theta}, \boldsymbol{x}_n)$ for per-sample losses and predictions. For gradients, we write $\boldsymbol{g}_n(\boldsymbol{\theta}) = \nabla_{\boldsymbol{\theta}} \ell_n(\boldsymbol{\theta})$ and $\boldsymbol{g}(\boldsymbol{\theta}) = \nabla_{\boldsymbol{\theta}} \mathcal{L}(\boldsymbol{\theta})$, suppressing $\boldsymbol{\theta}$ if unambiguous. We also set $\Theta = \mathbb{R}^D$ and $\mathbb{Y} = \mathbb{R}^C$ with $D, C$ the model parameter and prediction space dimension, respectively. For classification, $C$ is the number of classes.

**Hessian & GGN:** Two-fold chain rule application to the split $\ell \circ f$ decomposes the Hessian of Equation (1) into two parts $\nabla_{\boldsymbol{\theta}}^2 \mathcal{L}(\boldsymbol{\theta}) = \boldsymbol{G}(\boldsymbol{\theta}) + \boldsymbol{R}(\boldsymbol{\theta}) \in \mathbb{R}^{D \times D}$; the positive semi-definite GGN

$$\boldsymbol{G} = \tfrac{1}{N} \sum_{n=1}^{N} (\mathrm{J}_{\boldsymbol{\theta}} f_n)^\top \left(\nabla_{f_n}^2 \ell_n\right) (\mathrm{J}_{\boldsymbol{\theta}} f_n) = \tfrac{1}{N} \sum_{n=1}^{N} \boldsymbol{G}_n \tag{2}$$

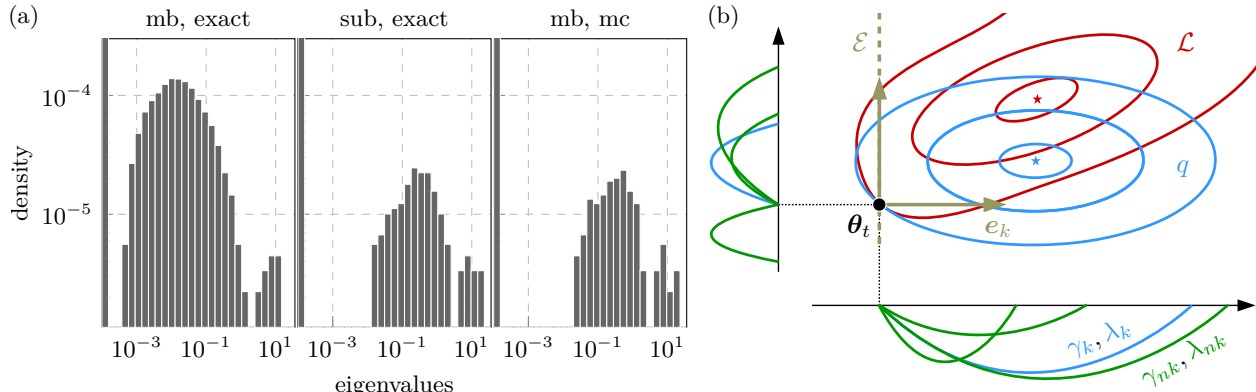

Figure 1: **Overview of ViViT's quantities:** **(a)** GGN eigenvalue distribution of DeepOBS' 3c3d architecture on CIFAR-10 (Schneider et al., 2019) for settings with different costs on a mini-batch of size $N = 128$. From left to right: Exact GGN, exact GGN on a mini-batch fraction, MC approximation of the GGN. **(b)** Pictorial illustration of ViViT's quantities in the loss landscape: the contour lines visualize the loss function $\mathcal{L}$ (Equation (1)) and its quadratic model $q$ around $\boldsymbol{\theta}_t \in \mathbb{R}^2$ (Equation (6)). The low-rank structure provides efficient access to the GGN's eigenvectors $\{\boldsymbol{e}_k\}$. The quadratic model's one-dimensional projections along the eigenvectors (black axes) are parabolas characterized by the directional derivatives $\gamma_k, \lambda_k$ and their per-sample contributions $\gamma_{nk}, \lambda_{nk}$ (Equation (8)). $\mathcal{E}$ is the GGN's top-1 eigenspace.

and a residual $\boldsymbol{R} = 1/N \sum_{n=1}^N \sum_{c=1}^C (\nabla_{\boldsymbol{\theta}}^2 [f_n]_c) [\nabla_{f_n} \ell_n]_c$. Here, we use the Jacobian $\mathrm{J}_{\boldsymbol{a}} \boldsymbol{b}$ that contains partial derivatives of $\boldsymbol{b}$ with respect to $\boldsymbol{a}$, $[\mathrm{J}_{\boldsymbol{a}} \boldsymbol{b}]_{ij} = \partial [\boldsymbol{b}]_i / \partial [\boldsymbol{a}]_j$. As the residual may alter the Hessian's definiteness – undesirable in many applications – we focus on the GGN.

**Low-rank structure:**  By basic inequalities, Equation (2) has rank$(\boldsymbol{G}) \le NC$.[1] To make this explicit, we factorize the positive semi-definite Hessian $\nabla_{f_n}^2 \ell_n = \sum_{c=1}^C \boldsymbol{s}_{nc} \boldsymbol{s}_{nc}^\top$, where $\boldsymbol{s}_{nc} \in \mathbb{R}^C$ and denote its backpropagated version by $\boldsymbol{v}_{nc} = (\mathrm{J}_{\boldsymbol{\theta}} f_n)^\top \boldsymbol{s}_{nc} \in \mathbb{R}^D$. Absorbing sums into matrix multiplications, we arrive at the GGN's outer product form that lies at the heart of the ViViT concept,

$$\boldsymbol{G} = \tfrac{1}{N} \sum_{n=1}^N \sum_{c=1}^C \boldsymbol{v}_{nc} \boldsymbol{v}_{nc}^\top = \boldsymbol{V} \boldsymbol{V}^\top \quad \text{with} \quad \boldsymbol{V} = \tfrac{1}{\sqrt{N}} (\boldsymbol{v}_{11}, \boldsymbol{v}_{12}, \dots, \boldsymbol{v}_{NC}) \in \mathbb{R}^{D \times NC}. \tag{3}$$

$\boldsymbol{V}$ allows for *exact* computations with the explicit GGN matrix, at linear rather than quadratic memory cost in $D$. We first formulate the extraction of relevant GGN properties from this factorization, before addressing how to further approximate $\boldsymbol{V}$ to reduce memory and computation costs.

### 2.1 Computing the full GGN eigenspectrum

Each GGN eigenvalue $\lambda \in \mathbb{R}_{\ge 0}$ is a root of the characteristic polynomial $\det(\boldsymbol{G} - \lambda \boldsymbol{I}_D)$ with identity matrix $\boldsymbol{I}_D \in \mathbb{R}^{D \times D}$. Leveraging the factorization of Equation (3) and the matrix determinant lemma, the $D$-dimensional eigenproblem reduces to that of the much smaller Gram matrix $\tilde{\mathbf{G}} = \boldsymbol{V}^\top \boldsymbol{V} \in \mathbb{R}^{NC \times NC}$ which contains pairwise scalar products of $\boldsymbol{v}_{nc}$ (see Appendix A.1),

$$\det(\boldsymbol{G} - \lambda \boldsymbol{I}_D) = 0 \quad \Leftrightarrow \quad \det(\tilde{\mathbf{G}} - \lambda \boldsymbol{I}_{NC}) = 0. \tag{4}$$

With at least $D - NC$ trivial solutions, the GGN curvature is zero along most directions in parameter space. Nontrivial solutions that give rise to curved directions are fully-contained in the Gram matrix, and hence *much* cheaper to compute.

Despite various Hessian spectral studies which rely on iterative eigensolvers and implicit matrix multiplication (Sagun et al., 2017; 2018; Adams et al., 2018; Ghorbani et al., 2019; Papyan, 2019b; Yao et al., 2019; Granziol et al., 2021), we are not aware of works that efficiently extract the *exact* GGN spectrum from its Gram

---

[1] We assume the overparameterized deep learning setting ($NC < D$) and suppress the trivial rank bound $D$.

matrix. In contrast to those techniques, this matrix can be computed in parallel with gradients in a single backward pass, which results in less sequential overhead. We demonstrate in Section 3.1 that exploiting the low-rank structure for computing the leading eigenpairs is superior to a power iteration based on matrix-free multiplication in terms of runtime.

Eigenvalues themselves can help identify reasonable hyperparameters, like learning rates (LeCun et al., 1993). But we can also reconstruct the associated eigenvectors. These are directions along which curvature information is contained in the mini-batch. Let $\tilde{\mathbb{S}}_+ = \{(\lambda_k, \tilde{\mathbf{e}}_k) \mid \lambda_k \neq 0, \tilde{\mathbf{G}}\tilde{\mathbf{e}}_k = \lambda_k \tilde{\mathbf{e}}_k\}_{k=1}^K$ denote the nontrivial Gram spectrum[2] with orthonormal eigenvectors $\tilde{\mathbf{e}}_j^\top \tilde{\mathbf{e}}_k = \delta_{jk}$ ($\delta$ represents the Kronecker delta and $K = \mathrm{rank}(\boldsymbol{G})$). Then, the transformed vectors $\boldsymbol{e}_k = {}^1\!/\!{\sqrt{\lambda_k}} \boldsymbol{V}\tilde{\mathbf{e}}_k$ ($k = 1, ..., K$) are orthonormal eigenvectors of $\boldsymbol{G}$ associated to eigenvalues $\lambda_k$ (see Appendix A.2), i.e.

$$\forall(\lambda_k, \tilde{\mathbf{e}}_k) \in \tilde{\mathbb{S}}_+: \quad \tilde{\mathbf{G}}\tilde{\mathbf{e}}_k = \lambda_k \tilde{\mathbf{e}}_k \implies \boldsymbol{G}\boldsymbol{e}_k = \lambda_k \boldsymbol{e}_k. \tag{5}$$

The eigenspectrum also provides access to the GGN's pseudo-inverse based on $\boldsymbol{V}$ and $\tilde{\mathbb{S}}_+$, required by e.g. second-order methods.[3]

## 2.2 Computing directional derivatives

Various algorithms rely on a local quadratic approximation of the loss landscape. For instance, optimization methods adapt their parameters by stepping into the minimum of the local proxy. Curvature, in the form of the Hessian or GGN, allows to build a quadratic model given by the Taylor expansion. Let $q$ denote the quadratic model for the loss around position $\boldsymbol{\theta}_t \in \Theta$ that uses curvature represented by the GGN,

$$q(\boldsymbol{\theta}) = \mathrm{const} + (\boldsymbol{\theta} - \boldsymbol{\theta}_t)^\top \boldsymbol{g}(\boldsymbol{\theta}_t) + \tfrac{1}{2}(\boldsymbol{\theta} - \boldsymbol{\theta}_t)^\top \boldsymbol{G}(\boldsymbol{\theta}_t)(\boldsymbol{\theta} - \boldsymbol{\theta}_t). \tag{6}$$

At its base point $\boldsymbol{\theta}_t$, the shape of $q$ along an arbitrary normalized direction $\boldsymbol{e} \in \Theta$ (i.e. $\|\boldsymbol{e}\| = 1$) is determined by the local gradient and curvature. Specifically, the projection of Equation (6) onto $\boldsymbol{e}$ gives rise to the (scalar) first-and second-order directional derivatives

$$\gamma_{\boldsymbol{e}} = \boldsymbol{e}^\top \nabla_{\boldsymbol{\theta}} q(\boldsymbol{\theta}_t) = \boldsymbol{e}^\top \boldsymbol{g}(\boldsymbol{\theta}_t) \in \mathbb{R} \qquad \text{and} \qquad \lambda_{\boldsymbol{e}} = \boldsymbol{e}^\top \nabla_{\boldsymbol{\theta}}^2 q(\boldsymbol{\theta}_t)\, \boldsymbol{e} = \boldsymbol{e}^\top \boldsymbol{G}(\boldsymbol{\theta}_t)\, \boldsymbol{e} \in \mathbb{R}. \tag{7}$$

As $\boldsymbol{G}$'s characteristic directions are its eigenvectors, they form a natural basis for the quadratic model. Denoting $\gamma_k = \gamma_{\boldsymbol{e}_k}$ and $\lambda_k = \lambda_{\boldsymbol{e}_k}$ the directional gradient and curvature along eigenvector $\boldsymbol{e}_k$, we see from Equation (7) that the directional curvature indeed coincides with the GGN's eigenvalue.

Analogous to the gradient and GGN, the directional derivatives $\gamma_k$ and $\lambda_k$ inherit the loss function's sum structure (Equation (1)), i.e. they decompose into contributions from individual samples. Let $\gamma_{nk}$ and $\lambda_{nk}$ denote these first- and second-order derivatives contributions of sample $\boldsymbol{x}_n$ in direction $k$,

$$\gamma_{nk} = \boldsymbol{e}_k^\top \boldsymbol{g}_n = \frac{\tilde{\mathbf{e}}_k^\top \boldsymbol{V}^\top \boldsymbol{g}_n}{\sqrt{\lambda_k}} \qquad \text{and} \qquad \lambda_{nk} = \boldsymbol{e}_k^\top \boldsymbol{G}_n \boldsymbol{e}_k = \frac{\|\boldsymbol{V}_n^\top \boldsymbol{V}\tilde{\mathbf{e}}_k\|^2}{\lambda_k}, \tag{8}$$

where $\boldsymbol{V}_n \in \mathbb{R}^{D \times C}$ is a scaled sub-matrix of $\boldsymbol{V}$ with fixed sample index. Note that directional derivatives can be evaluated efficiently with the Gram matrix eigenvectors without explicit access to the associated directions in parameter space.

In Equation (7), gradient $\boldsymbol{g}$ and curvature $\boldsymbol{G}$ are sums over $\boldsymbol{g}_n$ and $\boldsymbol{G}_n$, respectively. This implies the relationships between directional derivatives and per-sample contributions $\gamma_k = {}^1\!/\!{N} \sum_{n=1}^N \gamma_{nk}$ and $\lambda_k = {}^1\!/\!{N} \sum_{n=1}^N \lambda_{nk}$. Figure 1b shows a pictorial view of the quantities provided by VIVIT. Access to per-sample directional gradients $\gamma_{nk}$ and curvatures $\lambda_{nk}$ along $\boldsymbol{G}$'s natural directions is one distinct feature of VIVIT. These quantities provide geometric information about the local loss landscape *as well as* about the model's directional curvature stochasticity over the mini-batch.

---

[2]In the following, we assume ordered eigenvalues, i.e. $\lambda_1 \geq \lambda_2 \geq \ldots \geq \lambda_K$, for convenience.

[3]Appendix C.2 describes implicit multiplication with $\boldsymbol{G}^{-1}$.

## 2.3 Computational complexity

So far, we have formulated the computation of the GGN's eigenvalues (Equation (4)), including eigenvectors (Equation (5)), and per-sample directional derivatives (Equation (8)). We now analyze their computational complexity to identify critical performance factors. Those limitations can effectively be addressed with approximations that allow the costs to be decreased in a fine-grained fashion. We substantiate our theoretical analysis with empirical benchmarks in Section 3.1.

**Relation to gradient computation:**   Machine learning libraries are optimized to backpropagate signals $^1/_N \nabla_{f_n} \ell_n$ and accumulate the result into the mini-batch gradient $\boldsymbol{g} = ^1/_N \sum_{n=1}^N (J_{\boldsymbol{\theta}} f_n)^\top \nabla_{f_n} \ell_n$. Each column $\boldsymbol{v}_{nc}$ of $\boldsymbol{V}$ also involves applying the Jacobian, but to a different vector $\boldsymbol{s}_{nc}$ from the loss Hessian's symmetric factorization. For popular loss functions, like square and cross-entropy loss, this factorization is analytically known and available at negligible overhead. Hence, computing $\boldsymbol{V}$ basically costs $C$ gradient computations as it involves $NC$ backpropagations, while the gradient requires $N$. However, the practical overhead is expected to be smaller: Computations can re-use information from BackPACK's vectorized Jacobians and enjoy additional speedup on GPUs.

**Stage-wise discarding $\boldsymbol{V}$:**   The columns of $\boldsymbol{V}$ correspond to backpropagated vectors. During backpropagation, sub-matrices of $\boldsymbol{V}$, associated to parameters in the current layer, become available once at a time and can be discarded immediately after their use. This allows for memory savings without any approximations.

One example is the Gram matrix $\tilde{\mathbf{G}}$ formed by pairwise scalar products of $\{\boldsymbol{v}_{nc}\}_{n=1,c=1}^{N,C}$ in $\mathcal{O}((NC)^2 D)$ operations. The spectral decomposition $\tilde{\mathbb{S}}_+$ has additional cost of $\mathcal{O}((NC)^3)$. Similarly, the terms for the directional derivatives in Equation (8) can be built up stage-wise: First-order derivatives $\{\gamma_{nk}\}_{n=1,k=1}^{N,K}$ require the vectors $\{\boldsymbol{V}^\top \boldsymbol{g}_n \in \mathbb{R}^{NC}\}_{n=1}^N$ that cost $\mathcal{O}(N^2 CD)$ operations. Second-order derivatives are basically for free, as $\{\boldsymbol{V}_n^\top \boldsymbol{V} \in \mathbb{R}^{C \times NC}\}_{n=1}^N$ is available from $\tilde{\mathbf{G}}$.

**GGN eigenvectors:**   Transforming a Gram matrix eigenvector $\tilde{\mathbf{e}}_k$ to the GGN eigenvector $\boldsymbol{e}_k$ by application of $\boldsymbol{V}$ (Equation (5)) costs $\mathcal{O}(NCD)$ operations. However, repeated application of $\boldsymbol{V}$ can be avoided for sums of the form $\sum_k (^{c_k}/\sqrt{\lambda_k}) \boldsymbol{e}_k$ with arbitrary weights $c_k \in \mathbb{R}$. The summation can be performed in the Gram space at negligible overhead, and only the resulting vector $\sum_k c_k \tilde{\mathbf{e}}_k$ needs to be transformed. For a practical example – computing damped Newton steps – see Appendix B.1.

## 2.4 Approximations & Implementation

Although the GGN's representation by $\boldsymbol{V}$ has linear memory cost in $D$, it requires memory equivalent to $NC$ model copies.[4] Of course, this is infeasible for many networks and data sets, e.g. ImageNet ($C = 1000$). So far, our formulation was concerned with *exact* computations. We now present approximations that allow $N$, $C$ and $D$ in the above cost analysis to be replaced by smaller numbers, enabling ViViT to trade-off accuracy and performance.

**MC approximation & curvature sub-sampling:**   To reduce the scaling in $C$, we can approximate the factorization $\nabla_{f_n}^2 \ell_n(\boldsymbol{\theta}) = \sum_{c=1}^C \boldsymbol{s}_{nc} \boldsymbol{s}_{nc}^\top$ by a smaller set of vectors. One principled approach is to draw MC samples $\{\tilde{\mathbf{s}}_{nm}\}$ such that $\mathbb{E}_m[\tilde{\mathbf{s}}_{nm} \tilde{\mathbf{s}}_{nm}^\top] = \nabla_{f_n}^2 \ell_n(\boldsymbol{\theta})$ as in Dangel et al. (2020). This reduces the scaling of backpropagated vectors from $C$ to the number of MC samples $M$ ($M = 1$ in the following if not specified otherwise). A common independent approximation to reduce the scaling in $N$ is computing curvature on a mini-batch subset (Byrd et al., 2011; Zhang et al., 2017).

**Parameter groups (block-diagonal approximation):**   Some applications, e.g. computing Newton steps, require $\boldsymbol{V}$ to be kept in memory for performing the transformation from Gram space into the parameter

---

[4]Our implementation uses a more memory-efficient approach that avoids expanding $\boldsymbol{V}$ for linear layers by leveraging structure in their Jacobian. We describe additional optimizations in Appendix C.1, and demonstrate a 50x speed-up for computing the Gram matrix over naive computation via vectorized Jacobians. The additional backpropagations are carried out in 50 % of the expected time due to parallelism on the GPU.

space. Still, we can reduce costs by using the GGN's diagonal blocks $\{\boldsymbol{G}^{(i)}\}_{i=1}^{L}$ of each layer, rather than the full matrix $\boldsymbol{G}$. Such blocks are available during backpropagation and can thus be used and discarded step by step. In addition to the previously described approximations for reducing the costs in $N$ and $C$, this technique tackles scaling in $D$.

**Implementation details:** BACKPACK's functionality allows us to efficiently compute individual gradients and $\boldsymbol{V}$ in a single backward pass, using either an exact or MC-factorization of the loss Hessian. To reduce memory consumption, we extend its implementation with a protocol to support mini-batch sub-sampling and parameter groups. By hooks into the package's extensions, we can discard buffers as soon as possible during backpropagation, effectively implementing all discussed approximations and optimizations.

Next, we specifically address how the above approximations affect runtime and memory requirements, and study their impact on structural properties of the GGN.

## 3    Experiments

For the practical use of the VIVIT concept, it is essential that (i) the computations are efficient and (ii) that we gain an understanding of how sub-sampling noise and the approximations introduced in Section 2.4 alter the structural properties of the GGN. In the following, we therefore empirically investigate VIVIT's scalability and approximation properties in the context of deep learning, where it can serve as a monitoring tool of novel quantities that have not been explored previously to analyze training and other phenomena. The code used for the experiments is available at `https://github.com/f-dangel/vivit-experiments`.

**Experimental setting:** Architectures include three deep convolutional neural networks from DEEPOBS (Schneider et al., 2019) (2C2D on FASHION-MNIST, 3C3D on CIFAR-10 and ALL-CNN-C on CIFAR-100), as well as residual networks from He et al. (2016) on CIFAR-10 based on Idelbayev (2018) – all are equipped with cross-entropy loss. Based on the approximations presented in Section 2.4, we distinguish the following cases:

- **mb, exact:** Exact GGN with all mini-batch samples. Backpropagates $NC$ vectors.

- **mb, mc:** MC-approximated GGN with all mini-batch samples. Backpropagates $NM$ vectors with $M$ the number of MC-samples.

- **sub, exact:** Exact GGN on a subset of mini-batch samples ($\lfloor N/8 \rfloor$ as in Zhang et al. (2017)). Backpropagates $\lfloor N/8 \rfloor C$ vectors.

- **sub, mc:** MC-approximated GGN on a subset of mini-batch samples. Backpropagates $\lfloor N/8 \rfloor M$ vectors with $M$ the number of MC-samples.

### 3.1    Scalability

We now complement the theoretical computational complexity analysis from Section 2.3 with empirical studies. Results were generated on a workstation with an Intel Core i7-8700K CPU (32 GB) and one NVIDIA GeForce RTX 2080 Ti GPU (11 GB). We use $M = 1$ in the following.

**Memory performance:** We consider two tasks:

1. **Computing eigenvalues:** The nontrivial eigenvalues $\{\lambda_k \,|\, (\lambda_k, \tilde{\mathbf{e}}_k) \in \tilde{\mathbb{S}}_+\}$ are obtained by forming and eigen-decomposing the Gram matrix $\tilde{\mathbf{G}}$, allowing stage-wise discarding of $\boldsymbol{V}$ (see Sections 2.1 and 2.3).

2. **Computing the top eigenpair:** For $(\lambda_1, \boldsymbol{e}_1)$, we compute the Gram matrix spectrum $\tilde{\mathbb{S}}_+$, extract its top eigenpair $(\lambda_1, \tilde{\mathbf{e}}_1)$, and transform it into parameter space by Equation (5), i.e. $(\lambda_1, \boldsymbol{e}_1 = 1/\sqrt{\lambda_1} \boldsymbol{V} \tilde{\mathbf{e}}_1)$. This requires more memory than task 1 as $\boldsymbol{V}$ must be stored.

As a comprehensive measure for memory performance, we use the largest batch size before our system runs out of memory – we call this the critical batch size $N_{\text{crit}}$.

Figure 2a tabularizes the critical batch sizes on GPU for the 3C3D architecture on CIFAR-10. As expected, computing eigenpairs requires more memory and leads to consistently smaller critical batch sizes in comparison to computing only eigenvalues. Yet, they all exceed the traditional batch size used for training ($N = 128$, see Schneider et al. (2019)), even when using the exact GGN. With VIVIT's approximations, the memory overhead is reduced to significantly increase the applicable batch size. We report similar results for more architectures, a block-diagonal approximation (as in Zhang et al. (2017)), and on CPU in Appendix B.1, where we also benchmark a third task – computing damped Newton steps.

In the large-batch regime, computing the Gram matrix and its spectrum becomes a run time bottleneck, see Section 2.3. As we show next, VIVIT is highly efficient in the mini-batch setting. However, for spectral analyses on the entire dataset, iterative eigensolvers, like power iterations, should be preferred.

**Runtime performance:** Here, we consider the task of computing the $k$ leading eigenvectors and eigenvalues of a matrix. A power iteration that computes eigenpairs iteratively via matrix-vector products serves as a reference. For a fixed value of $k$, we repeat both approaches 20 times and report the shortest time.

For the power iteration, we adapt the implementation from the PYHESSIAN library (Yao et al., 2019) and replace its Hessian-vector product by a matrix-free GGN-vector product (Schraudolph, 2002) through PYTORCH's automatic differentiation. We use the same default hyperparameters for the termination criterion.[5] Similar to task 1, our method obtains the top-$k$ eigenpairs[6] by computing $\tilde{\mathbb{S}}_+$, extracting its leading eigenpairs and transforming the eigenvectors $\tilde{\mathbf{e}}_1, \tilde{\mathbf{e}}_2, \ldots, \tilde{\mathbf{e}}_k$ into parameter space by application of $\boldsymbol{V}$.

Figure 2b shows the GPU runtime for the 3C3D architecture on CIFAR-10, using a mini-batch of size $N = 128$. Without any approximations to the GGN, our method already outperforms the power iteration for $k > 1$ and increases *much* slower in run time as more leading eigenpairs are requested. This means that, relative to the transformation of each eigenvector from the Gram space into the parameter space through $\boldsymbol{V}$, the run time mainly results from computing $\boldsymbol{V}, \tilde{\mathbf{G}}$, and eigendecomposing the latter. This is consistent with the computational complexity of those operations in $NC$ (compare Section 2.3) and allows for efficient extraction of a large number of eigenpairs. The run time curves of the approximations confirm this behavior by featuring the same flat profile. Additionally, they require significantly less time than the exact mini-batch computation. Appendix B.1 reports additional results for more architectures, a block-diagonal approximation, and on CPU.

## 3.2 Approximation quality

VIVIT is based on the Hessian's generalized Gauss-Newton approximation (see Equation (2)). In practice, the GGN is only computed on a mini-batch which yields a statistical estimator for the *full-batch* GGN (i.e. the GGN evaluated on the entire training set). Additionally, we introduce curvature sub-sampling and an MC approximation (see Section 2.4), i.e. further approximations that alter the curvature's structural properties. In this section, we compare quantities at different stages within this hierarchy of approximations. We use the test problems from above and train the networks with both SGD and ADAM (details in Appendix B.2).

**GGN vs. Hessian:** First, we empirically study the relationship between the GGN and the Hessian in the deep learning context. To capture *solely* the effect of neglecting the residual $\boldsymbol{R}$ (see Equation (2)), we consider the noise-free case and compute $\boldsymbol{H}$ and $\boldsymbol{G}$ on the entire training set.

We characterize both curvature matrices by their top-$C$ eigenspace: the space spanned by the $C$ leading eigenvectors. This is a $C$-dimensional subspace of the parameter space $\Theta$, on which the loss function is subject to particularly strong curvature. The *overlap* between these spaces serves as the comparison metric. Let $\{\boldsymbol{e}_c^U\}_{c=1}^C$ be the set of orthonormal eigenvectors to the $C$ largest eigenvalues of some symmetric matrix $\boldsymbol{U}$ and $\mathcal{E}^U = \text{span}(\boldsymbol{e}_1^U, ..., \boldsymbol{e}_C^U)$. The projection onto this subspace $\mathcal{E}^U$ is given by the projection matrix

---

[5]We find similar results with relaxed convergence hyperparameters, see Appendix B.1.
[6]The power iteration computes the leading eigenpairs. Our approach allows choosing arbitrary eigenpairs.

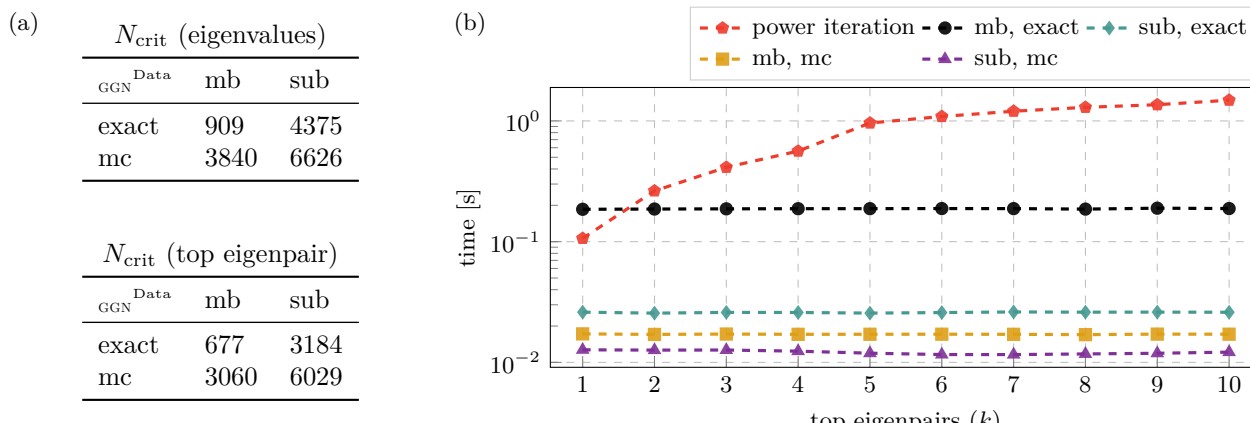

Figure 2: **GPU memory and run time performance:** Performance measurements for the 3C3D architecture ($D = 895{,}210$) on CIFAR-10 ($C = 10$). **(a)** Critical batch sizes $N_{\text{crit}}$ for computing eigenvalues and the top eigenpair. **(b)** Run time comparison with a power iteration for extracting the $k$ leading eigenpairs using a batch of size $N = 128$.

$\boldsymbol{P}^{\boldsymbol{U}} = (\boldsymbol{e}_1^{\boldsymbol{U}}, ..., \boldsymbol{e}_C^{\boldsymbol{U}})(\boldsymbol{e}_1^{\boldsymbol{U}}, ..., \boldsymbol{e}_C^{\boldsymbol{U}})^\top$. As in Gur-Ari et al. (2018), we define the overlap between two top-$C$ eigenspaces $\mathcal{E}^{\boldsymbol{U}}$ and $\mathcal{E}^{\boldsymbol{V}}$ of the matrices $\boldsymbol{U}$ and $\boldsymbol{V}$ by

$$\text{overlap}(\mathcal{E}^{\boldsymbol{U}}, \mathcal{E}^{\boldsymbol{V}}) = \frac{\text{Tr}\left(\boldsymbol{P}^{\boldsymbol{U}} \boldsymbol{P}^{\boldsymbol{V}}\right)}{\sqrt{\text{Tr}\left(\boldsymbol{P}^{\boldsymbol{U}}\right) \text{Tr}\left(\boldsymbol{P}^{\boldsymbol{V}}\right)}} \in [0, 1]. \tag{9}$$

If $\text{overlap}(\mathcal{E}^{\boldsymbol{U}}, \mathcal{E}^{\boldsymbol{V}}) = 0$, then $\mathcal{E}^{\boldsymbol{U}}$ and $\mathcal{E}^{\boldsymbol{V}}$ are orthogonal; if the overlap is 1, they are identical.

Figure 3a shows the overlap between the full-batch GGN and Hessian during training of the 3C3D network on CIFAR-10 with SGD. Except for a short phase at the beginning of the optimization procedure (note the log scale for the epoch-axis), it shows a strong agreement (overlap $\geq 0.85$) between the top-$C$ eigenspaces. We make similar observations with the other test problems (see Appendix B.3), yet to a slightly lesser extent for CIFAR-100. Consequently, we identify the GGN as an interesting object, since it consistently shares relevant structure with the Hessian matrix.

**Eigenspace under noise and approximations:** VIVIT uses mini-batching to compute a statistical estimator of the full-batch GGN. This approximation alters the top-$C$ eigenspace, as shown in Figure 3b: With decreasing mini-batch size, the approximation carries less and less structure of its full-batch counterpart, as indicated by dropping overlaps. In addition, at constant batch size, a decrease in approximation quality can be observed over the course of training. This might be a valuable insight for the design of second-order optimization methods, where this structural decay could lead to performance degradation over the course of the optimization, which has to be compensated for by a growing batch-size (e.g. Martens (2010) reports that the optimal batch size grows during training).

In order to allow for a fine-grained cost-accuracy trade-off, VIVIT introduces *further* approximations to the mini-batch GGN (see Section 2.4). Figure 3c shows the overlap between these GGN approximations and the full-batch GGN.[7] The order of the approximations is as expected: With increasing computational effort, the approximations improve and, despite the greatly reduced computational effort compared to the exact mini-batch GGN, significant structure of the top-$C$ eigenspace is preserved. Details and results for the other test problems are reported in Appendix B.4.

So far, our analysis is based on the top-$C$ eigenspace of the curvature matrices. We extend this analysis by studying the effect of noise and approximations on the curvature *magnitude* along the top-$C$ directions in Appendix B.5.

---

[7]A comparison with the mini-batch GGN as ground truth can be found in Appendix B.4

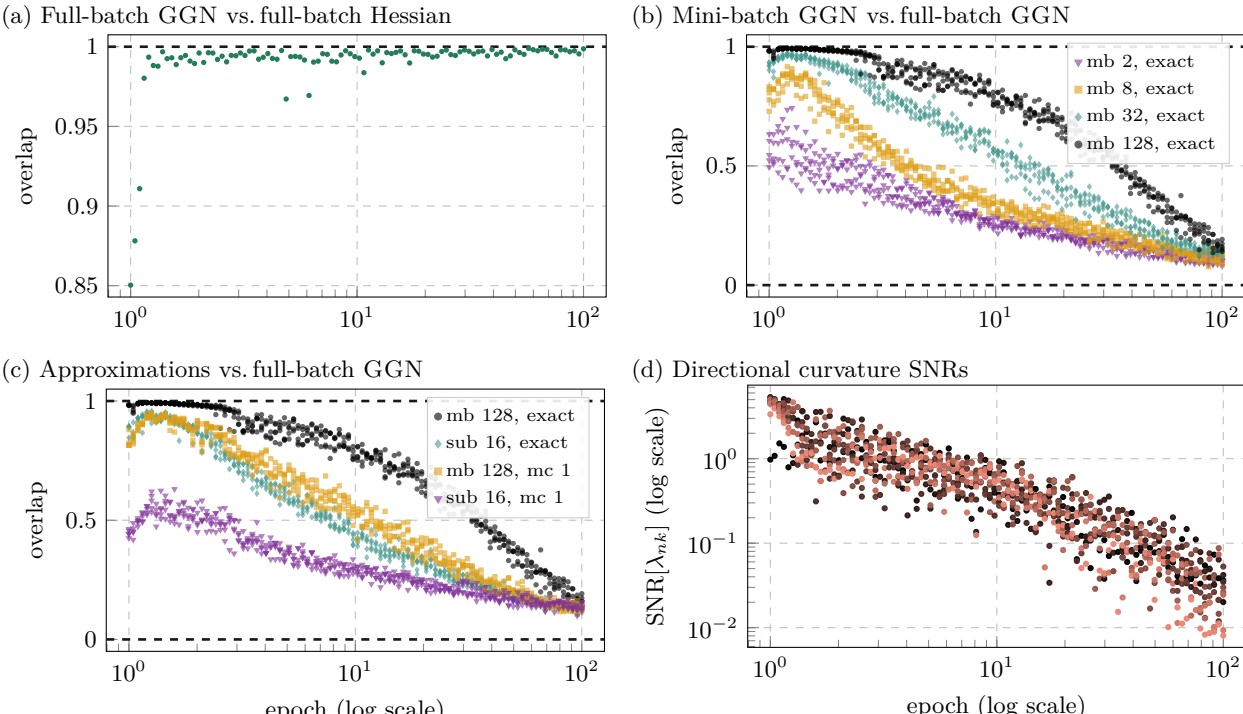

Figure 3: **Curvature monitoring during training 3c3d on CIFAR-10 with SGD:** **(a)** Overlap between the top-$C$ eigenspaces of the full-batch GGN and full-batch Hessian during training. **(b)** Overlap between the top-$C$ eigenspaces of the mini-batch GGN and full-batch GGN during training. For each mini-batch size, 5 different mini-batches are drawn. **(c)** Overlap between the top-$C$ eigenspaces of the mini-batch GGN and VIVIT's approximations with the full-batch GGN during training. Each approximation is evaluated on 5 mini-batches. **(d)** Curvature SNRs along each of the mini-batch GGN's top-$C$ eigenvectors during training. At fixed epoch, the SNR for the most curved direction is shown in ● and the SNR for the direction with the smallest curvature is shown in ●.

### 3.3 Per-sample directional derivatives

A unique feature of VIVIT's quantities is that they provide a notion of *curvature uncertainty* through *per-sample* first- and second-order directional derivatives (see Equation (8)). To quantify noise in the directional derivatives, we compute their signal-to-noise ratios (SNRs). For each direction $\boldsymbol{e}_k$, the SNR is given by the squared empirical mean divided by the empirical variance of the $N$ mini-batch samples $\{\gamma_{nk}\}_{n=1}^{N}$ and $\{\lambda_{nk}\}_{n=1}^{N}$, respectively.

Figure 3d shows curvature SNRs during training the 3c3d network on CIFAR-10 with SGD. The curvature signal along the top-$C$ eigenvectors decreases from SNR $> 1$ by two orders of magnitude. This might be a challenging setting for second-order methods because the noise varies dramatically during different stages of training. In comparison, the directional gradients do not exhibit such a pattern (see Appendix B.6). Results for the other test cases can be found in Appendix B.6.

In this section, we gave a glimpse of the *very rich* quantities that are efficiently computed via VIVIT. Section 5 discusses the practical use of this information, in particular curvature uncertainty.

## 4 Related work

**GGN spectrum & low-rank structure:** Other works point out the GGN's low-rank structure. Botev et al. (2017) present the rank bound and propose an alternative to K-FAC based on backpropagating a

decomposition of the loss Hessian. Papyan (2019a) presents the factorization in Equation (3) and studies the eigenvalue spectrum's hierarchy for cross-entropy loss. In this setting, the GGN further decomposes into summands, some of which are then analyzed through similar Gram matrices. These can be obtained as contractions of $\tilde{\mathbf{G}}$, but our approach goes beyond them as it does not neglect terms. We are not aware of works that obtain the exact spectrum *and* leverage a highly-efficient fully-parallel implementation. This may be because, until recently (Bradbury et al., 2020; Dangel et al., 2020), vectorized Jacobians required to perform those operations efficiently were not available.

**Efficient operations with large-scale matrices in deep learning:** Chen et al. (2021) use Equation (3) for element-wise evaluation of the GGN in fully-connected feed-forward neural networks. They also present a variant based on MC sampling. This element-wise evaluation is then used to construct hierarchical matrix approximations of the GGN. VIVIT instead leverages the global low-rank structure that also enjoys efficient eigen-decomposition.

A special case of VIVIT's Gram matrix extraction is computing empirical neural tangent kernel (NTK) matrices, like Novak et al. (2022). While the NTK only requires a model (its Jacobian (Jacot et al., 2018)), the GGN also incorporates the loss function via its Hessian. For mean squared error, this Hessian in Equation (2) is proportional to the identity, and the GGN Gram matrix coincides with the scaled empirical NTK.

Another prominent low-rank matrix in deep learning is the un-centered gradient covariance (sometimes called empirical Fisher). Singh & Alistarh (2020) describe implicit multiplication with its inverse and apply it for neural network compression, assuming the empirical Fisher as Hessian proxy. However, this assumption has limitations, specifically for optimization (Kunstner et al., 2019). In principle though, the low-rank structure also permits the application of our methods from Section 2.

## 5 Use cases

Our efficient implementation enables the community to explore deep learning through richer information that would previously have been costly. Here, we want to briefly address possible use cases – their full development and assessment, however, will amount to separate paper(s). They include:

- **Second-order optimization:** Second-order methods use curvature to build a local quadratic model of the loss. Established curvature proxies neglect the sampling-induced noise and therefore the quadratic model's reliability. VIVIT provides access to this information in the form of *per-sample* quantities. This offers a new dimension for improving second-order methods: Through statistics on the mini-batch *distribution* of directional derivatives, we might be able to adapt to the dynamics of noise (e.g. via variance-adapted step sizes).

- **Monitoring tool:** However, to develop conceptually novel second-order optimizers, we believe that a crucial intermediate step is to better understand the setting they operate in. The techniques we present primarily tackle this intermediate step. Sections 3.2 and 3.3 are examples of VIVIT's application as a monitoring tool. Due to its efficiency, VIVIT could be integrated into live diagnostic tools like COCKPIT (Schneider et al., 2021) that aim at debugging optimizers or gaining insights into the inner workings of deep learning.

## 6 Conclusion

We have presented VIVIT, a curvature model based on the low-rank structure of the Hessian's generalized Gauss-Newton (GGN) approximation. This structure allows for efficient extraction of *exact* curvature properties, such as the GGN's full eigenvalue spectrum and directional gradients and curvatures along the associated eigenvectors. VIVIT's quantities scale by approximations that allow for a fine-grained cost-accuracy trade-off. In contrast to alternatives, these quantities offer a notion of curvature uncertainty across the mini-batch in the form of directional derivatives.

We empirically demonstrated the efficiency of leveraging the GGN's low-rank structure and substantiated its usefulness by studying characteristics of curvature noise on various deep learning architectures. The low-rank

representation is efficiently computed in parallel with gradients during a single backward pass. As it mainly relies on vectorized Jacobians, it is general enough to be integrated into existing machine learning libraries in the future. For now, we provide an efficient open-source implementation in PyTorch (Paszke et al., 2019) by extending the existing BackPACK (Dangel et al., 2020) library.

**Acknowledgments**

The authors gratefully acknowledge financial support by the European Research Council through ERC StG Action 757275 / PANAMA; the DFG Cluster of Excellence "Machine Learning - New Perspectives for Science", EXC 2064/1, project number 390727645; the German Federal Ministry of Education and Research (BMBF) through the Tübingen AI Center (FKZ: 01IS18039A); and funds from the Ministry of Science, Research and Arts of the State of Baden-Württemberg. Moreover, the authors thank the International Max Planck Research School for Intelligent Systems (IMPRS-IS) for supporting Felix Dangel and Lukas Tatzel. Further, we are grateful to Agustinus Kristiadi, Filip de Roos, Frank Schneider, Jonathan Wenger, and Marius Hobbhahn for providing feedback to the manuscript.

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

# ViViT: Curvature Access Through The Generalized Gauss-Newton's Low-Rank Structure (Supplementary Material)

## A  Mathematical details

### A.1  Reducing the GGN eigenvalue problem to the Gram matrix

For Equation (4), consider the left hand side of the GGN's characteristic polynomial $\det(\boldsymbol{G} - \lambda \boldsymbol{I}_D) = 0$. Inserting the ViViT factorization (Equation (3)) and using the matrix determinant lemma yields

$$
\begin{aligned}
\det(-\lambda \boldsymbol{I}_D + \boldsymbol{G}) &= \det\!\left(-\lambda \boldsymbol{I}_D + \boldsymbol{V}\boldsymbol{V}^\top\right) && \text{(Low-rank structure (3))} \\
&= \det\!\left(\boldsymbol{I}_{NC} + \boldsymbol{V}^\top(-\lambda \boldsymbol{I}_D)^{-1}\boldsymbol{V}\right)\det(-\lambda \boldsymbol{I}_D) && \text{(Matrix determinant lemma)} \\
&= \det\!\left(\boldsymbol{I}_{NC} - \frac{1}{\lambda}\boldsymbol{V}^\top\boldsymbol{V}\right)(-\lambda)^D && \\
&= \left(-\frac{1}{\lambda}\right)^{NC}\det\!\left(\boldsymbol{V}^\top\boldsymbol{V} - \lambda \boldsymbol{I}_{NC}\right)(-\lambda)^D && \\
&= (-\lambda)^{D-NC}\det\!\left(\tilde{\mathbf{G}} - \lambda \boldsymbol{I}_{NC}\right). && \text{(Gram matrix)}
\end{aligned}
$$

Setting the above expression to zero reveals that the GGN's spectrum decomposes into $D - NC$ zero eigenvalues and the Gram matrix spectrum obtained from $\det(\tilde{\mathbf{G}} - \lambda \boldsymbol{I}_{NC}) = 0$.

### A.2 Relation between GGN and Gram matrix eigenvectors

Assume the nontrivial Gram matrix spectrum $\tilde{\mathbb{S}}_+ = \{(\lambda_k, \tilde{\mathbf{e}}_k) \mid \lambda_k \neq 0, \tilde{\mathbf{G}}\tilde{\mathbf{e}}_k = \lambda_k\tilde{\mathbf{e}}_k\}_{k=1}^K$ with orthonormal eigenvectors $\tilde{\mathbf{e}}_j^\top \tilde{\mathbf{e}}_k = \delta_{jk}$ ($\delta$ represents the Kronecker delta) and $K = \mathrm{rank}(\boldsymbol{G})$. We now show that $\boldsymbol{e}_k = {}^1\!/\!\sqrt{\lambda_k}\boldsymbol{V}\tilde{\mathbf{e}}_k$ are normalized eigenvectors of $\boldsymbol{G}$ and inherit orthogonality from $\tilde{\mathbf{e}}_k$.

To see the first, consider right-multiplication of the GGN with $\boldsymbol{e}_k$, then expand the low-rank structure,

$$\begin{aligned}
\boldsymbol{G}\boldsymbol{e}_k &= \frac{1}{\sqrt{\lambda_k}}\boldsymbol{V}\boldsymbol{V}^\top\boldsymbol{V}\tilde{\mathbf{e}}_k && \text{(Equation (3) and definition of } \boldsymbol{e}_k\text{)} \\
&= \frac{1}{\sqrt{\lambda_k}}\boldsymbol{V}\tilde{\mathbf{G}}\tilde{\mathbf{e}}_k && \text{(Gram matrix)} \\
&= \lambda_k\frac{1}{\sqrt{\lambda_k}}\boldsymbol{V}\tilde{\mathbf{e}}_k && \text{(Eigenvector property of } \tilde{\mathbf{e}}_k\text{)} \\
&= \lambda_k\boldsymbol{e}_k \,.
\end{aligned}$$

Orthonormality of the $\boldsymbol{e}_k$ results from the Gram matrix eigenvector orthonormality,

$$\begin{aligned}
\boldsymbol{e}_j^\top \boldsymbol{e}_k &= \left(\frac{1}{\sqrt{\lambda_j}}\tilde{\mathbf{e}}_j^\top\boldsymbol{V}^\top\right)\left(\frac{1}{\sqrt{\lambda_k}}\boldsymbol{V}\tilde{\mathbf{e}}_k\right) && \text{(Definition of } \boldsymbol{e}_j, \boldsymbol{e}_k\text{)} \\
&= \frac{1}{\sqrt{\lambda_j\lambda_k}}\tilde{\mathbf{e}}_j^\top\tilde{\mathbf{G}}\tilde{\mathbf{e}}_k && \text{(Gram matrix)} \\
&= \frac{\lambda_k}{\sqrt{\lambda_j\lambda_k}}\tilde{\mathbf{e}}_j^\top\tilde{\mathbf{e}}_k && \text{(Eigenvector property of } \tilde{\mathbf{e}}_k\text{)} \\
&= \delta_{jk} \,. && \text{(Orthonormality)}
\end{aligned}$$

## B Experimental details

Throughout this section, we use the notation introduced in Section 3 (see Table S.1). The code used for the experiments is available at https://github.com/f-dangel/vivit-experiments.

Table S.1: **Notation for curvature approximations:** The notation is introduced in Section 3. This table recapitulates the abbreviations (referring to the approximations introduced in Section 2.4) and provides corresponding explanations.

| Abbreviation | Explanation |
| --- | --- |
| **mb, exact** | Exact GGN with all mini-batch samples. |
| | Backpropagates $NC$ vectors. |
| **mb, mc** | MC-approximated GGN with all mini-batch samples. |
| | Backpropagates $NM$ vectors with $M$ the number of MC-samples. |
| **sub, exact** | Exact GGN on a subset of mini-batch samples ($\lfloor N/8 \rfloor$ as in Zhang et al. (2017)). |
| | Backpropagates $\lfloor N/8 \rfloor C$ vectors. |
| **sub, mc** | MC-approximated GGN on a subset of mini-batch samples. |
| | Backpropagates $\lfloor N/8 \rfloor M$ vectors with $M$ the number of MC-samples. |

**GGN spectra (Figure 1a):** To obtain the spectra of Figure 1a we initialize the respective architecture, then draw a mini-batch and evaluate the GGN eigenvalues under the described approximations, clipping the Gram matrix eigenvalues at $10^{-4}$. Figures S.4 and S.5 provide the spectra for all used architectures with both the full GGN and a per-layer block-diagonal approximation.

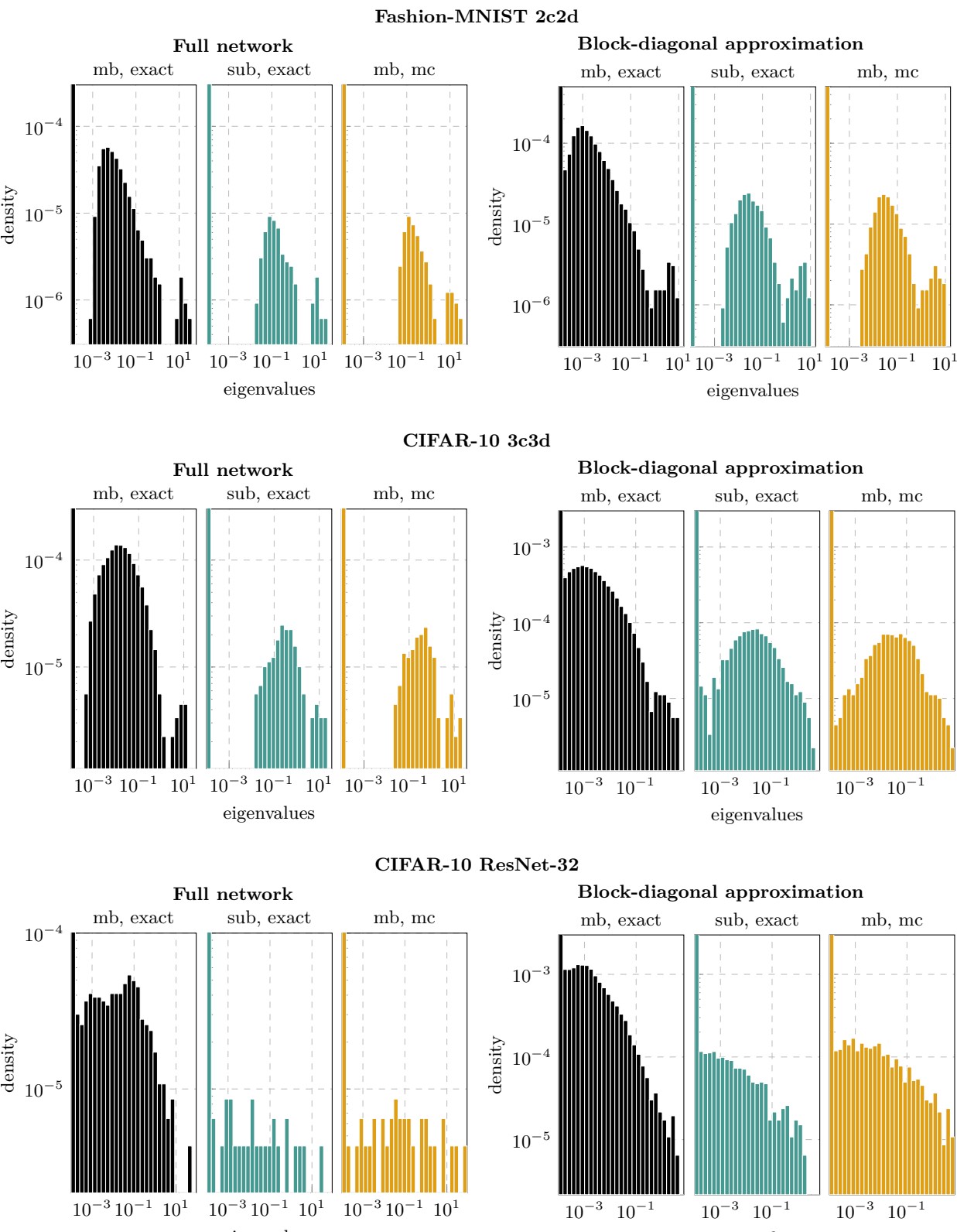

Figure S.4: **GGN spectra of different architectures under ViViT's approximations:** Left and right columns contain results with the full network's GGN and a per-layer block-diagonal approximation, respectively. The column labels are explained in Table S.1.

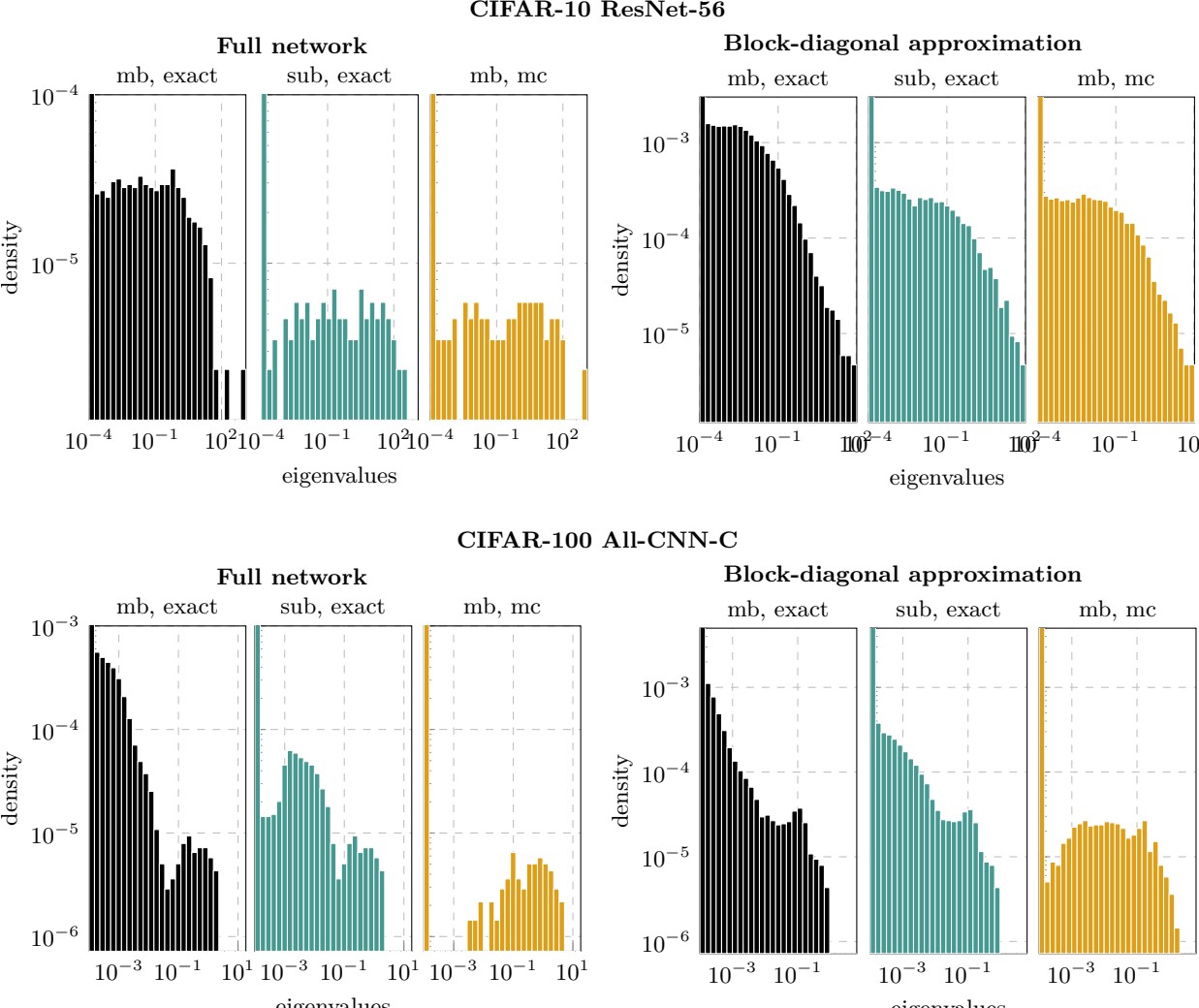

Figure S.5: **GGN spectra of different architectures under ViViT's approximations:** Left and right columns contain results with the full network's GGN and a per-layer block-diagonal approximation, respectively. The column labels are explained in Table S.1.

## B.1 Performance evaluation

**Hardware specifications:** Results in this section were generated on a workstation with an Intel Core i7-8700K CPU (32 GB) and one NVIDIA GeForce RTX 2080 Ti GPU (11 GB).

**Note:** ViViT's quantities are implemented through BACKPACK, which is triggered by PyTorch's gradient computation. Consequently, they can only be computed together with PyTorch's mini-batch gradient.

**Architectures:** We use untrained deep convolutional and residual networks from DeepOBS Schneider et al. (2019) and Idelbayev (2018). If a net has batch normalization layers, we set them to evaluation mode. Otherwise, the loss would not obey the sum structure of Equation (1). The batch normalization layers' internal moving averages, required for evaluation mode, are initialized by performing five forward passes with the current mini-batch in training mode before.

In experiments with fixed mini-batches the batch sizes correspond to DeepOBS' default value for training where possible (CIFAR-10: $N = 128$, Fashion-MNIST: $N = 128$). The residual networks use a batch size

of $N = 128$. On CIFAR-100 (trained with $N = 256$), we reduce the batch size to $N = 64$ to fit the exact computation on the full mini-batch, used as baseline, into memory. If the GGN approximation is evaluated on a subset of the mini-batch (**sub**), $\lfloor N/8 \rfloor$ of the samples are used (as in Zhang et al. (2017)). The MC approximation is always evaluated with a single sample ($M = 1$).

**Memory performance (critical batch sizes):** Two tasks are considered (see Section 3.1):

1. **Computing eigenvalues:** Compute the nontrivial eigenvalues $\{\lambda_k \,|\, (\lambda_k, \tilde{\mathbf{e}}_k) \in \tilde{\mathbb{S}}_+ \}$ .

2. **Computing the top eigenpair:** Compute the top eigenpair $(\lambda_1, \boldsymbol{e}_1)$.

We repeat the tasks above and vary the mini-batch size until the device runs out of memory. The largest mini-batch size that can be handled by our system is denoted as $N_{\mathrm{crit}}$, the critical batch size. We determine this number by bisection on the interval $[1; 32768]$.

Figures S.8 to S.17a,b present the results. As described in Section 2.3, computing eigenvalues is more memory-efficient than computing eigenvectors and exhibits larger critical batch sizes. In line with the description in Section 2.4, a block-diagonal approximation is usually more memory-efficient and results in a larger critical batch size. Curvature sub-sampling and MC approximation further increase the applicable batch sizes.

In summary, we find that there always exists a combination of approximations which allows for critical batch sizes larger than the traditional size used for training (some architectures even permit exact computation). Different accuracy-cost trade-offs may be preferred, depending on the application and the computational budget. By the presented approximations, VιVιT's representation is capable to adapt over a wide range.

**Runtime performance:** Here, we consider the task of computing the $k$ leading eigenvectors and eigenvalues of a matrix. VιVιT's eigenpair computation is compared with a power iteration that computes eigenpairs iteratively via matrix-vector products. The power iteration baseline is based on the PyHessian library Yao et al. (2019) and uses the same termination criterion (at most 100 matrix-vector products per eigenvalue; stop if the eigenvalue estimate's relative change is less than $10^{-3}$). In contrast to PyHessian, we use a different data format and stack the computed eigenvectors. This reduces the number of `for`-loops in the orthonormalization step. We repeat each run time measurement 20 times and report the shortest execution time as result.

Figures S.8 to S.17c,d show the results. For most architectures, our exact method outperforms the power iteration for $k > 1$ and increases only marginally in runtime as the number of requested eigenvectors grows. The proposed approximations share this property, and further reduce run time.

**Power iteration with relaxed hyperparameters:** When VιVιT's approximations are used for computing eigenvalues, a power iteration with relaxed hyperparameters might be an alternative. We thus extend the runtime experiment from Section 3.1 in the following way: We consider the task of computing the top-10 eigenpairs of the mini-batch GGN on CIFAR-10 3c3d, with the same batch size $N = 128$, on GPU using both VιVιT and the power iteration. The resulting approximations of the eigenvalues $\hat{\lambda}_k, k = 1, \ldots, 10$ are compared with the exact eigenvalues $\lambda_k, k = 1, \ldots, 10$ using

$$1 - \frac{1}{10} \sum_{k=1}^{10} \frac{|\hat{\lambda}_k - \lambda_k|}{|\lambda_k|}$$

as a measure of accuracy. We run the power iteration with 20 different convergence tolerances varying on a logarithmic grid from $10^{-5}$ to $10^{-1}$, and disable termination due to exceeded iterations. Each setting is repeated 20 times, and we report the best run time and corresponding accuracy in Figure S.6.

In this setting, computing the exact top-10 eigenvalues with VιVιT is faster than approximating them with a power iteration, even when using the largest tested tolerance. Also, when using VιVιT with approximations through sub-sampling and MC-approximation, these runs require less run time than the power iteration

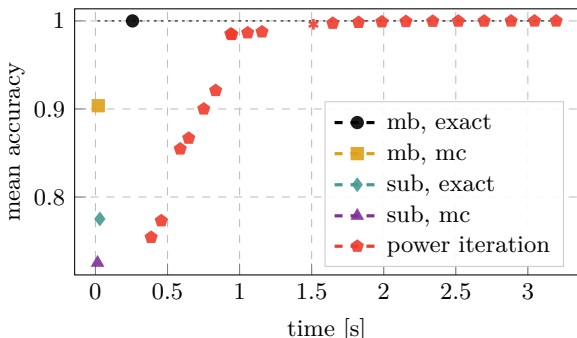

Figure S.6: **Accuracy per run time performance:** Comparison of ViViT and its approximations to the power iteration for computing the top-10 eigenvalues for the 3c3d architecture ($D = 895{,}210$) on CIFAR-10 ($C = 10$). For the power iteration, different tolerances ranging from $10^{-5}$ to $10^{-1}$ are used. The red star shows the result for the default tolerance.

at similar accuracy. In the plot, we also highlighted the power iteration run with PyHessian's default convergence parameters (red star marker). It seems to yield a relatively good trade-off between time and accuracy for the power iteration.

These results show that, also in terms of "accuracy per run time", ViViT is superior to a power iteration in the mini-batch setting due to its increased parallelism.

***Power iteration on the* GGN *versus power iteration on the* GGN *Gram matrix:*** In the run time evaluation of our method (Section 3.1) we compute the $k$ leading GGN eigenpairs by computing the full GGN Gram matrix spectrum and discarding all but the leading eigenpairs. Here, we present additional results where we exchange the full diagonalization by a power iteration with identical convergence hyperparameters as the baseline; a power iteration on the GGN.

Figure S.7 visualizes the comparison for the same setting as Figure 2b in the main text. In case of no approximations (mb, exact) where the Gram matrix dimension is largest, the power iteration can further reduce the run time shown in Figure 2b. However, for the GGN approximations through sub-sampling or MC approximation, the power iteration on the (rather small) Gram matrix, deteriorates performance in comparison to the results reported in Figure 2b as the number of leading eigenvalues increases. In this regime (small Gram matrix, many requested eigenvalues), the simplistic power iteration can require more matrix-vector products than a sophisticated eigensolver that computes the full spectrum. As in the main text, these findings show that ViViT (even in the exact version) outperforms the power iteration for $k \geq 2$.

**Note on CIFAR-100 (large C):** For data sets with a large number of classes, like CIFAR-100 ($C = 100$), computations with the exact GGN are costly. In particular, constructing the Gram matrix $\tilde{\mathbf{G}}$ has quadratic memory cost in $C$, and its eigendecomposition has cubic cost in time with $C$ (see Section 2.3).

As a result, the exact computation only works with batch sizes smaller than DeepOBS' default ($N = 256$ for CIFAR-100, see Figures S.16 and S.17a,b). For the GGN block-diagonal approximation, which fits into CPU memory for $N = 64$, the exact computation of top eigenpairs is slower than a power iteration and only becomes comparable if a large number of eigenpairs is requested, see Figure S.17d.

For such data sets, the approximations proposed in Section 2.4 are essential to reduce costs. The most effective approximation to eliminate the scaling with $C$ is using an MC approximation. Figures S.16 and S.17 confirm that the approximate computations scale to batch sizes used for training and that computing eigenpairs takes less time than a power iteration.

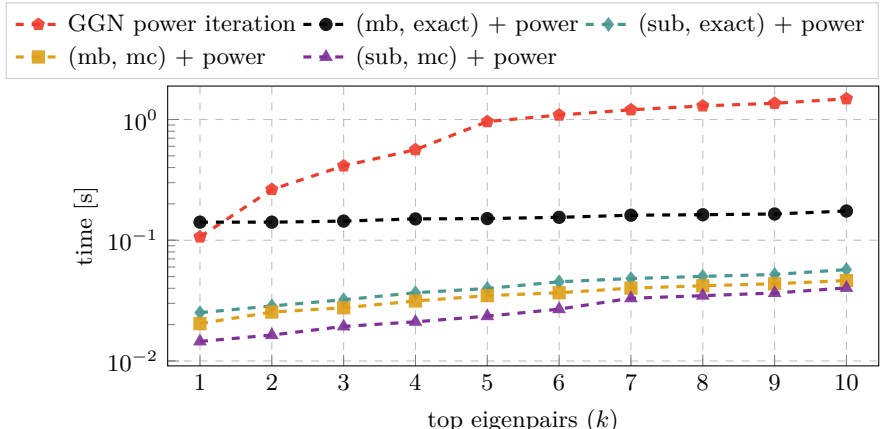

Figure S.7: **Power iteration on the GGN versus power iteration on the Gram matrix:** The figure considers the same setting as Figure 2b (3c3d on CIFAR-10 and GPU). However, we exchange the full diagonalization of the GGN's Gram matrix by a power iteration with identical convergence hyperparameters as the baseline power iteration on the GGN (at most 100 matrix-vector producst per eigenvalue; stop if the eigenvalue estimate's relative change is less than $10^{-3}$).

**Computing damped Newton steps:** A Newton step $-(\boldsymbol{G} + \delta\boldsymbol{I})^{-1}\boldsymbol{g}$ with damping $\delta > 0$ can be decomposed into updates along the eigenvectors of the GGN $\boldsymbol{G}$,

$$-(\boldsymbol{G} + \delta\boldsymbol{I})^{-1}\boldsymbol{g} = \sum_{k=1}^{K} \frac{-\gamma_k}{\lambda_k + \delta}\boldsymbol{e}_k + \sum_{k=K+1}^{D} \frac{-\gamma_k}{\delta}\boldsymbol{e}_k \,. \tag{S.10}$$

It corresponds to a Newton update along nontrivial eigendirections that uses the first- and second-order directional derivatives described in Section 2.2 and a gradient descent step with learning rate $1/\delta$ along trivial directions (with $\lambda_k = 0$). In the following, we refer to the first summand of Equation (S.10) as Newton step. As described in Section 2.3, we can perform the weighted sum in the Gram matrix space, rather than the parameter space, by computing

$$\sum_{k=1}^{K} \frac{-\gamma_k}{\lambda_k + \delta}\boldsymbol{e}_k = \sum_{k=1}^{K} \frac{-\gamma_k}{\lambda_k + \delta}\frac{1}{\sqrt{\lambda_k}}\boldsymbol{V}\tilde{\mathbf{e}}_k = \boldsymbol{V}\left(\sum_{k=1}^{K} \frac{-\gamma_k}{(\lambda_k + \delta)\sqrt{\lambda_k}}\tilde{\mathbf{e}}_k\right) \,.$$

This way, only a single vector needs to be transformed from Gram space into parameter space.

Table S.2 shows the critical batch sizes for the Newton step computation (first term on the right-hand side of Equation (S.10)), using Gram matrix eigenvalues larger than $10^{-4}$ and constant damping $\delta = 1$. Second-order directional derivatives $\lambda_k$ are evaluated on the same samples as the GGN eigenvectors, but we *always* use all mini-batch samples to compute the directional gradients $\gamma_n$. Using our approximations, the Newton step computation scales to batch sizes beyond the traditional sizes used for training.

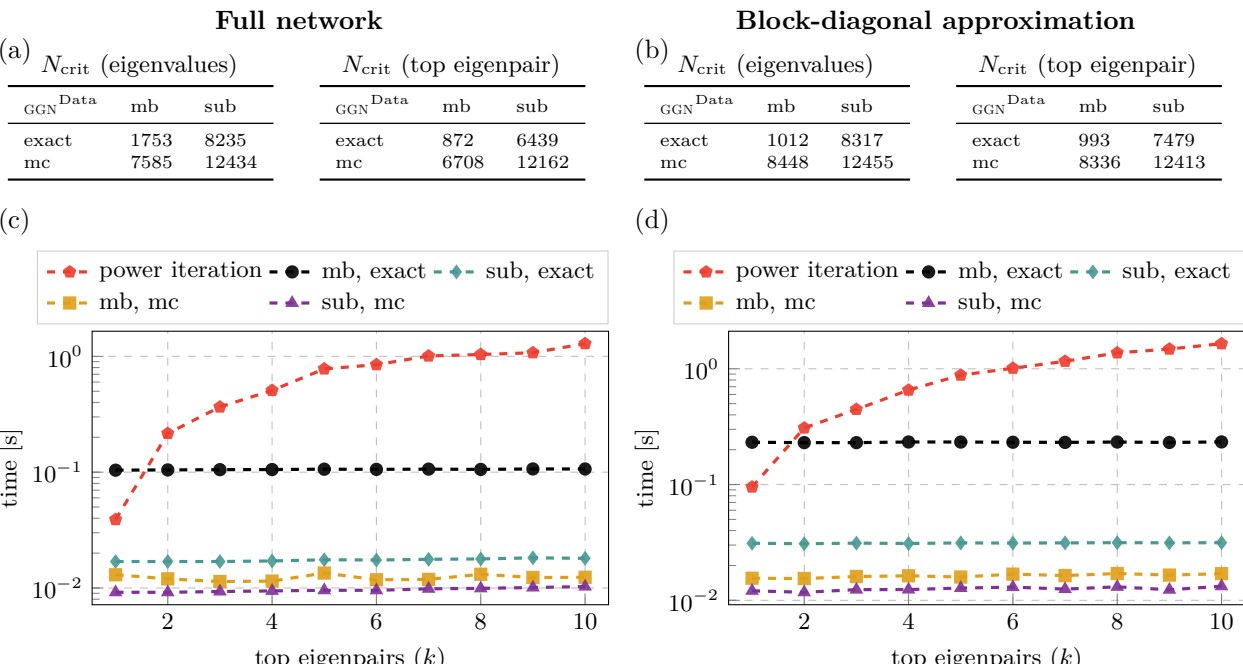

Figure S.8: **GPU memory and run time performance for the 2c2d architecture on Fashion-MNIST:** Left and right columns show results with the full network's GGN ($D = 3{,}274{,}634$, $C = 10$) and a per-layer block-diagonal approximation, respectively. (a, b) Critical batch sizes $N_{\mathrm{crit}}$ for computing eigenvalues and the top eigenpair. (c, d) Run time comparison with a power iteration for extracting the $k$ leading eigenpairs using a mini-batch of size $N = 128$.

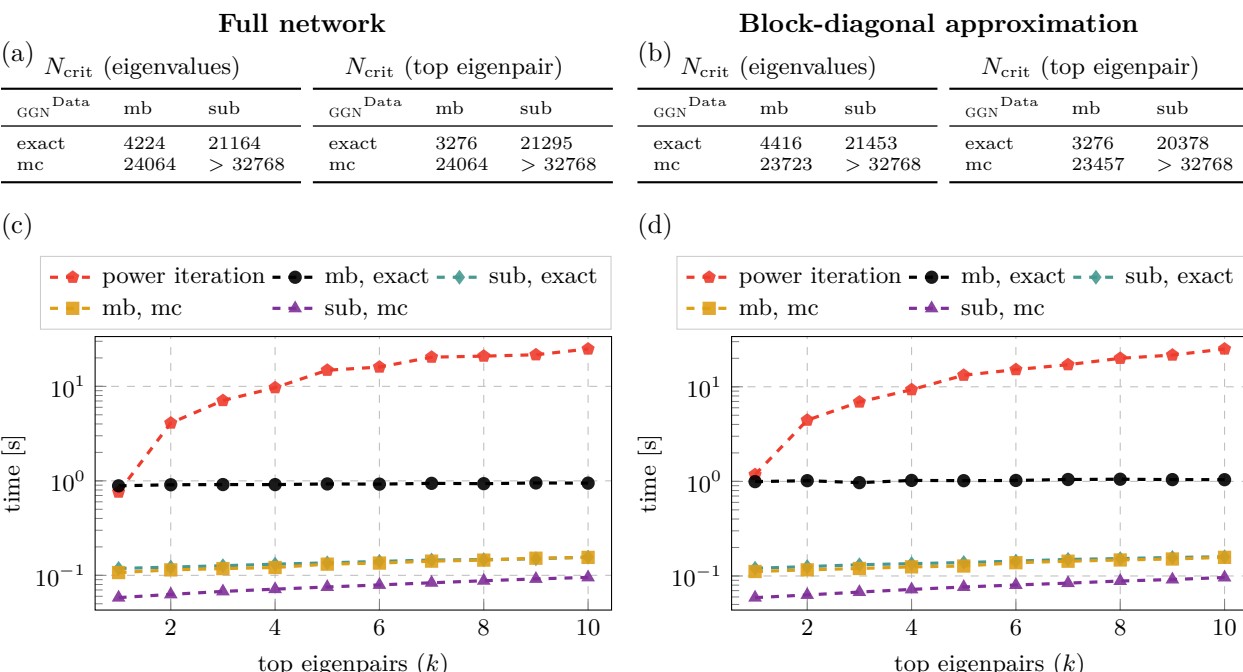

Figure S.9: **CPU memory and run time performance for the 2c2d architecture on Fashion-MNIST:** Left and right columns show results with the full network's GGN ($D = 3{,}274{,}634$, $C = 10$) and a per-layer block-diagonal approximation, respectively. (a, b) Critical batch sizes $N_{\mathrm{crit}}$ for computing eigenvalues and the top eigenpair. (c, d) Run time comparison with a power iteration for extracting the $k$ leading eigenpairs using a mini-batch of size $N = 128$.

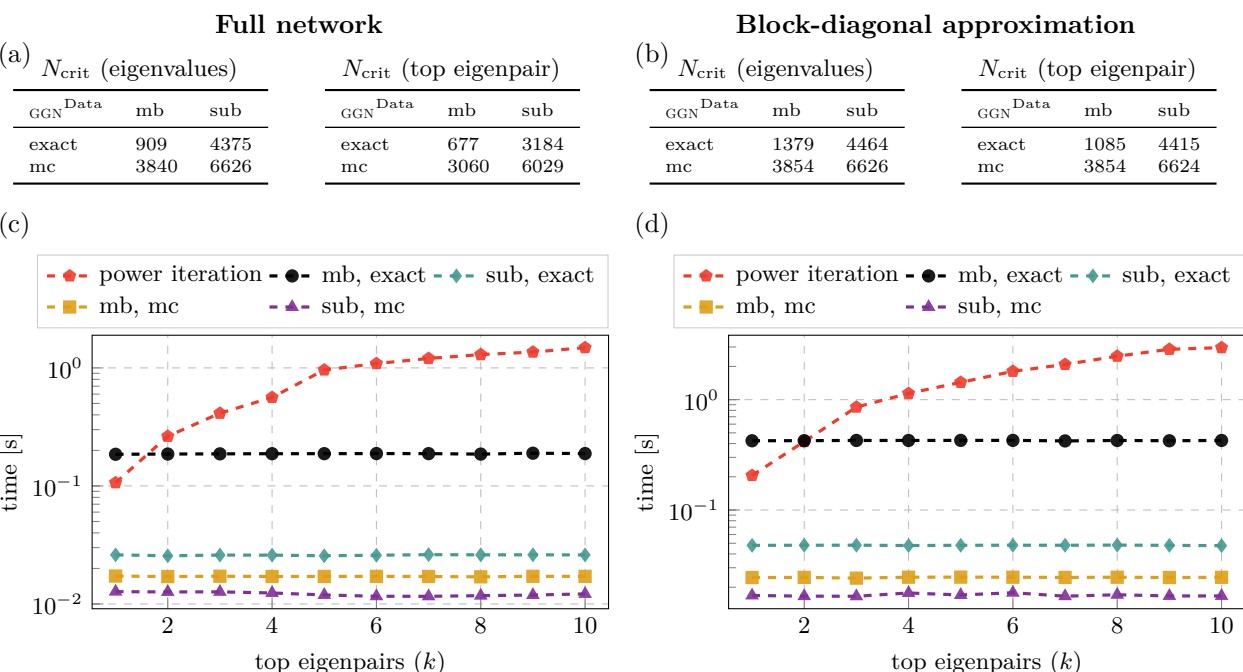

Figure S.10: **GPU memory and run time performance for the 3c3d architecture on CIFAR-10:** Left and right columns show results with the full network's GGN ($D = 895{,}210$, $C = 10$) and a per-layer block-diagonal approximation, respectively. **(a, b)** Critical batch sizes $N_{\mathrm{crit}}$ for computing eigenvalues and the top eigenpair. **(c, d)** Run time comparison with a power iteration for extracting the $k$ leading eigenpairs using a mini-batch of size $N = 128$.

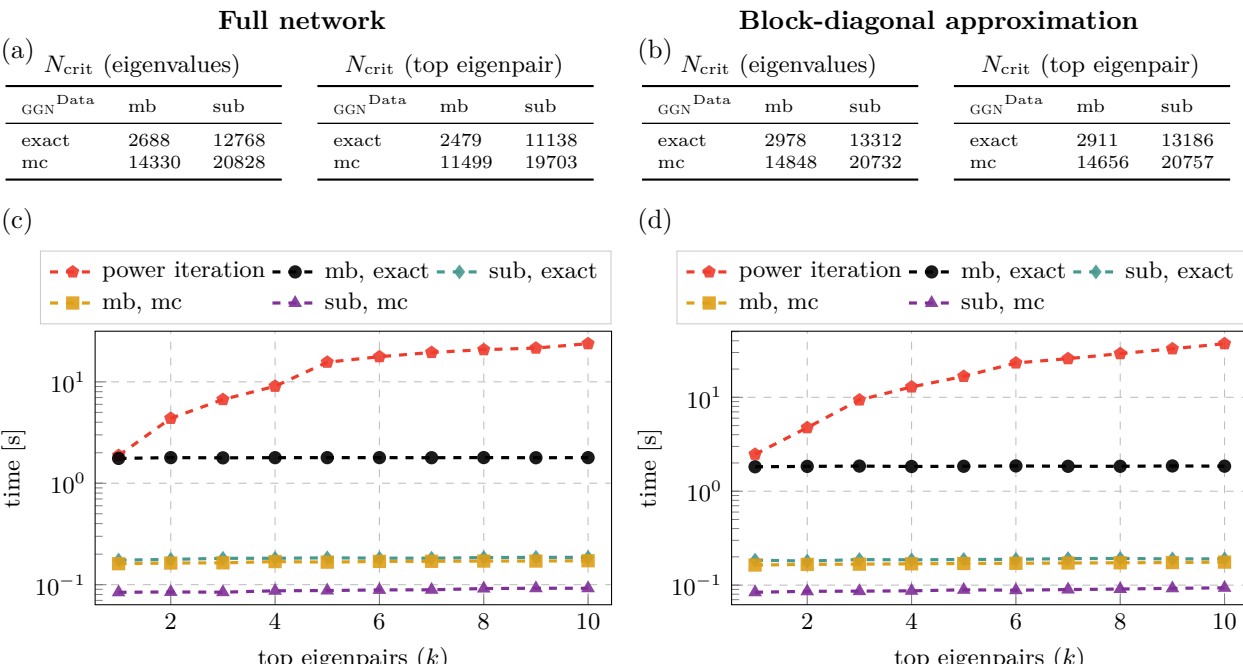

Figure S.11: **CPU memory and run time performance for the 3c3d architecture on CIFAR-10:** Left and right columns show results with the full network's GGN ($D = 895{,}210$, $C = 10$) and a per-layer block-diagonal approximation, respectively. (a, b) Critical batch sizes $N_{\text{crit}}$ for computing eigenvalues and the top eigenpair. (c, d) Run time comparison with a power iteration for extracting the $k$ leading eigenpairs using a mini-batch of size $N = 128$.

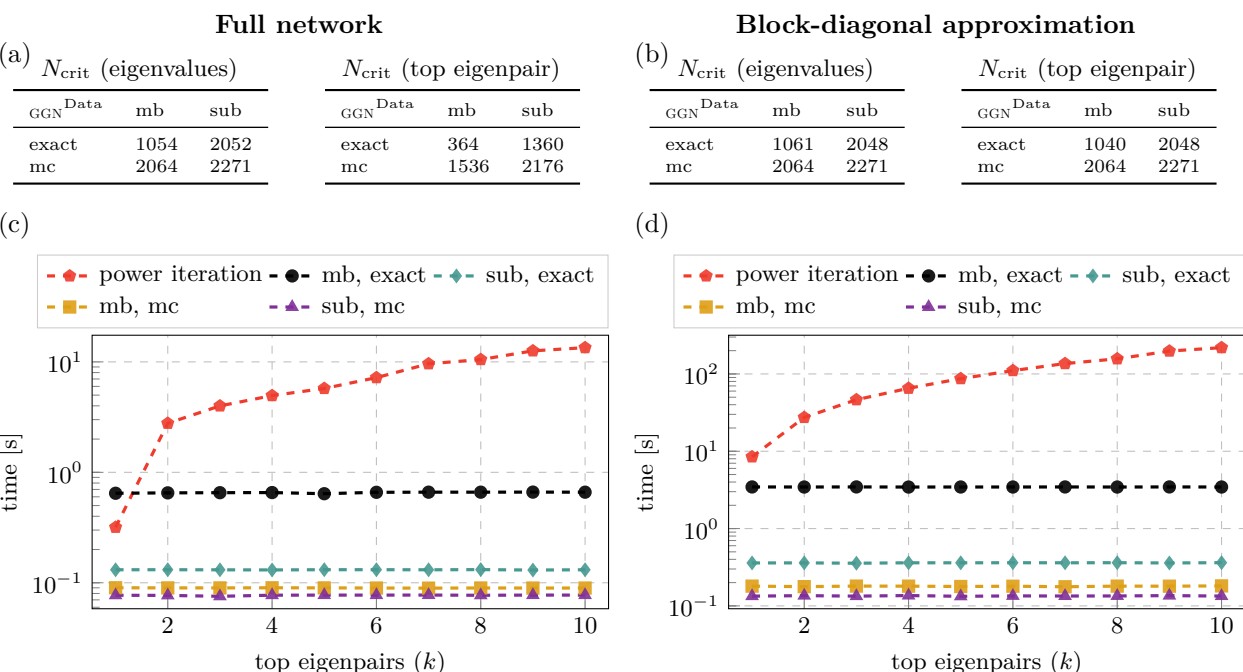

Figure S.12: **GPU memory and run time performance for the ResNet-32 architecture on CIFAR-10:** Left and right columns show results with the full network's GGN ($D = 464{,}154$, $C = 10$) and a per-layer block-diagonal approximation, respectively. (a, b) Critical batch sizes $N_{\mathrm{crit}}$ for computing eigenvalues and the top eigenpair. (c, d) Run time comparison with a power iteration for extracting the $k$ leading eigenpairs using a mini-batch of size $N = 128$.

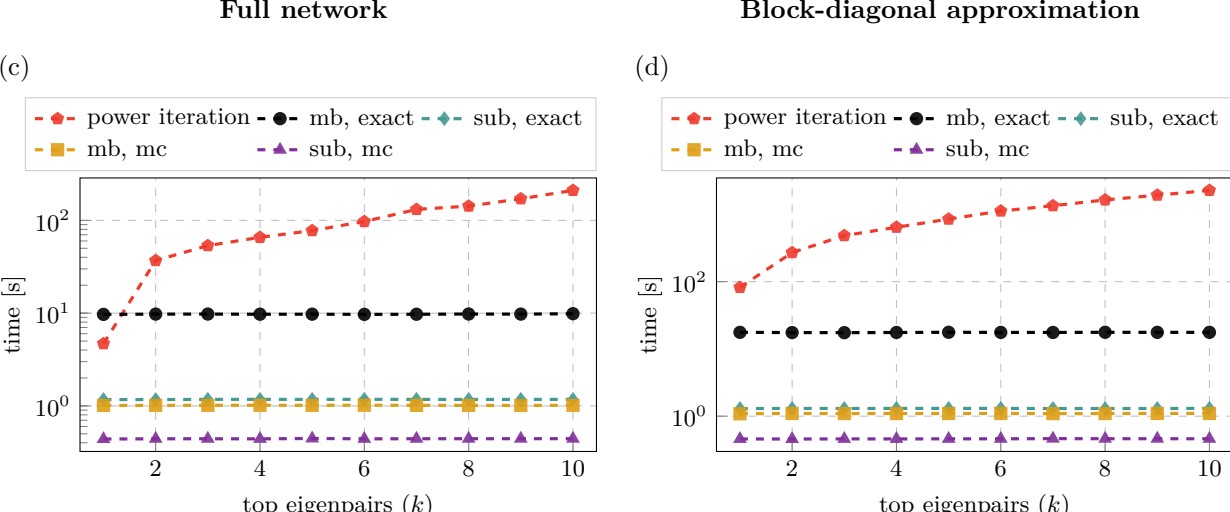

Figure S.13: **CPU memory and run time performance for the ResNet-32 architecture on CIFAR-10:** Left and right columns show results with the full network's GGN ($D = 464{,}154$, $C = 10$) and a per-layer block-diagonal approximation, respectively. (c, d) Run time comparison with a power iteration for extracting the $k$ leading eigenpairs using a mini-batch of size $N = 128$.

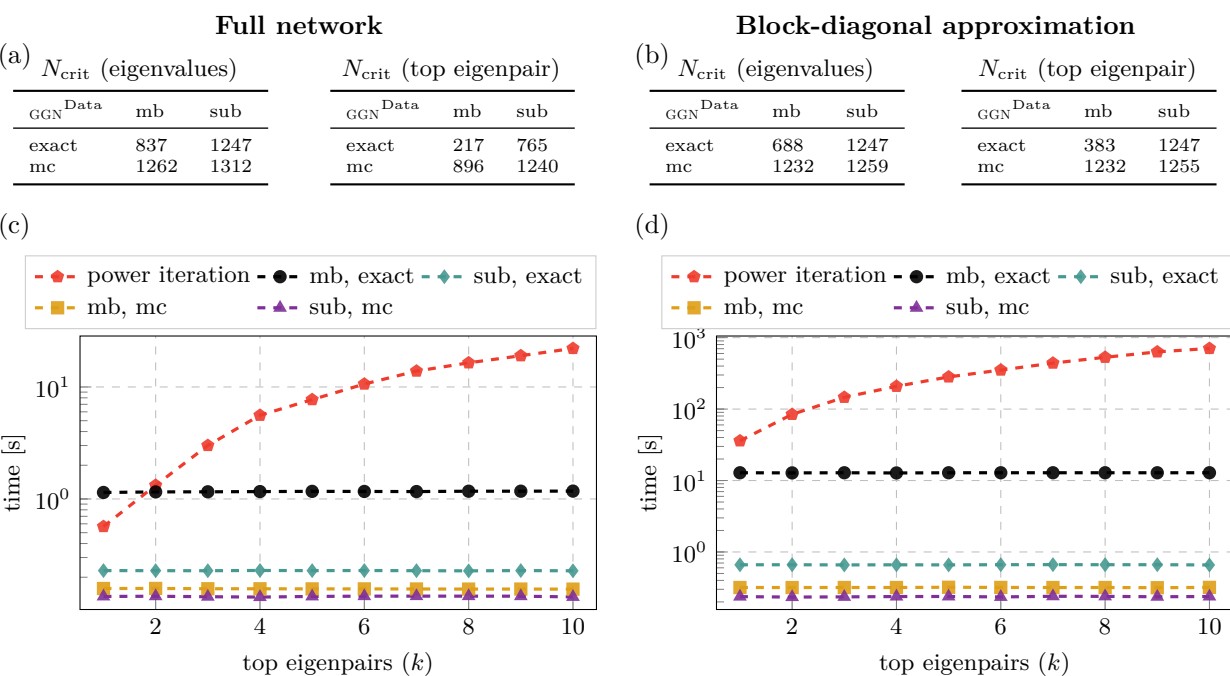

Figure S.14: **GPU memory and run time performance for the ResNet-56 architecture on CIFAR-10:** Left and right columns show results with the full network's GGN ($D = 853{,}018$, $C = 10$) and a per-layer block-diagonal approximation, respectively. (a, b) Critical batch sizes $N_{\mathrm{crit}}$ for computing eigenvalues and the top eigenpair. (c, d) Run time comparison with a power iteration for extracting the $k$ leading eigenpairs using a mini-batch of size $N = 128$.

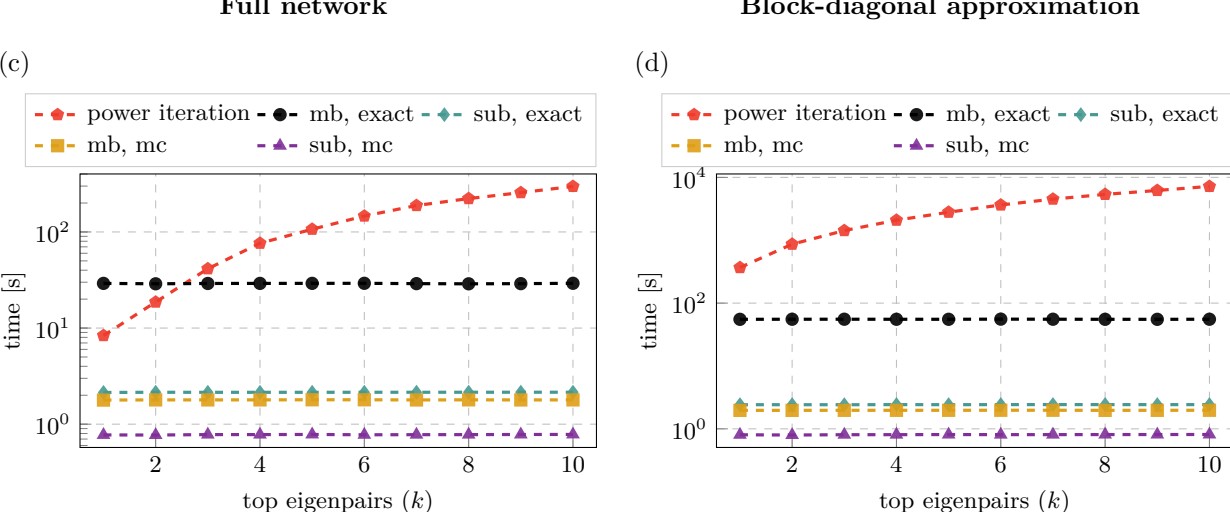

Figure S.15: **CPU memory and run time performance for the ResNet-56 architecture on CIFAR-10:** Left and right columns show results with the full network's GGN ($D = 853{,}018$, $C = 10$) and a per-layer block-diagonal approximation, respectively. (c, d) Run time comparison with a power iteration for extracting the $k$ leading eigenpairs using a mini-batch of size $N = 128$.

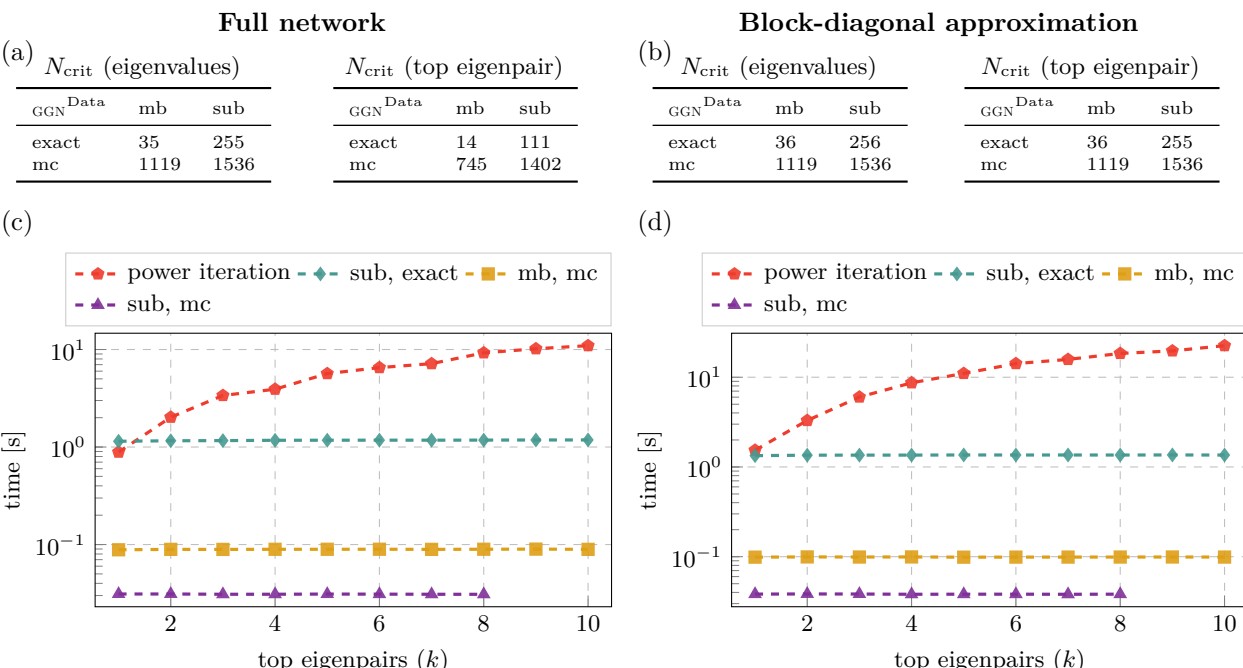

Figure S.16: **GPU memory and run time performance for the All-CNN-C architecture on CIFAR-100:** Left and right columns show results with the full network's GGN ($D = 1{,}387{,}108, C = 100$) and a per-layer block-diagonal approximation, respectively. (a, b) Critical batch sizes $N_{\mathrm{crit}}$ for computing eigenvalues and the top eigenpair. (c, d) Run time comparison with a power iteration for extracting the $k$ leading eigenpairs using a mini-batch of size $N = 64$.

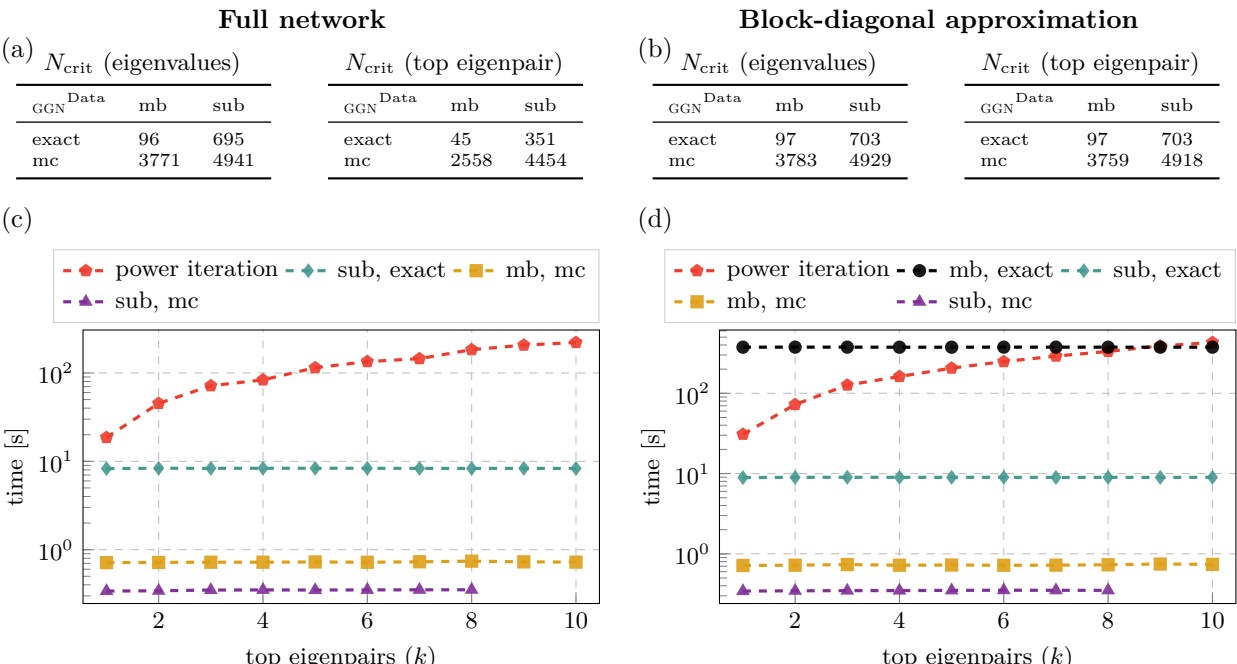

Figure S.17: **CPU memory and run time performance for the All-CNN-C architecture on CIFAR-100:** Left and right columns show results with the full network's GGN ($D = 1{,}387{,}108, C = 100$) and a per-layer block-diagonal approximation, respectively. (a, b) Critical batch sizes $N_{\mathrm{crit}}$ for computing eigenvalues and the top eigenpair. (c, d) Run time comparison with a power iteration for extracting the $k$ leading eigenpairs using a mini-batch of size $N = 64$.

Table S.2: **Memory performance for computing damped Newton steps:** Left and right columns show the critical batch sizes with the full network's GGN and a per-layer block-diagonal approximation, respectively.

**Fashion-MNIST 2c2d**

| Full network | | | | | | Block-diagonal approximation | | | | | |
|---|---|---|---|---|---|---|---|---|---|---|---|
| $N_{\mathrm{crit}}$ (GPU) | | | $N_{\mathrm{crit}}$ (CPU) | | | $N_{\mathrm{crit}}$ (GPU) | | | $N_{\mathrm{crit}}$ (CPU) | | |
| $\mathrm{GGN}^{\mathrm{Data}}$ | mb | sub | $\mathrm{GGN}^{\mathrm{Data}}$ | mb | sub | $\mathrm{GGN}^{\mathrm{Data}}$ | mb | sub | $\mathrm{GGN}^{\mathrm{Data}}$ | mb | sub |
| exact | 66 | 159 | exact | 202 | 487 | exact | 68 | 159 | exact | 210 | 487 |
| mc | 362 | 528 | mc | 1107 | 1639 | mc | 368 | 528 | mc | 1137 | 1643 |

**CIFAR-10 3c3d**

| Full network | | | | | | Block-diagonal approximation | | | | | |
|---|---|---|---|---|---|---|---|---|---|---|---|
| $N_{\mathrm{crit}}$ (GPU) | | | $N_{\mathrm{crit}}$ (CPU) | | | $N_{\mathrm{crit}}$ (GPU) | | | $N_{\mathrm{crit}}$ (CPU) | | |
| $\mathrm{GGN}^{\mathrm{Data}}$ | mb | sub | $\mathrm{GGN}^{\mathrm{Data}}$ | mb | sub | $\mathrm{GGN}^{\mathrm{Data}}$ | mb | sub | $\mathrm{GGN}^{\mathrm{Data}}$ | mb | sub |
| exact | 208 | 727 | exact | 667 | 2215 | exact | 349 | 795 | exact | 1046 | 2423 |
| mc | 1055 | 1816 | mc | 3473 | 5632 | mc | 1659 | 2112 | mc | 4997 | 6838 |

**CIFAR-10 ResNet-32**

| Full network | | | Block-diagonal approximation | | |
|---|---|---|---|---|---|
| $N_{\mathrm{crit}}$ (GPU) | | | $N_{\mathrm{crit}}$ (GPU) | | |
| $\mathrm{GGN}^{\mathrm{Data}}$ | mb | sub | $\mathrm{GGN}^{\mathrm{Data}}$ | mb | sub |
| exact | 344 | 1119 | exact | 1051 | 1851 |
| mc | 1205 | 1535 | mc | 2048 | 2208 |

**CIFAR-10 ResNet-56**

| Full network | | | Block-diagonal approximation | | |
|---|---|---|---|---|---|
| $N_{\mathrm{crit}}$ (GPU) | | | $N_{\mathrm{crit}}$ (GPU) | | |
| $\mathrm{GGN}^{\mathrm{Data}}$ | mb | sub | $\mathrm{GGN}^{\mathrm{Data}}$ | mb | sub |
| exact | 209 | 640 | exact | 767 | 1165 |
| mc | 687 | 890 | mc | 1232 | 1255 |

**CIFAR-100 All-CNN-C**

| Full network | | | | | | Block-diagonal approximation | | | | | |
|---|---|---|---|---|---|---|---|---|---|---|---|
| $N_{\mathrm{crit}}$ (GPU) | | | $N_{\mathrm{crit}}$ (CPU) | | | $N_{\mathrm{crit}}$ (GPU) | | | $N_{\mathrm{crit}}$ (CPU) | | |
| $\mathrm{GGN}^{\mathrm{Data}}$ | mb | sub | $\mathrm{GGN}^{\mathrm{Data}}$ | mb | sub | $\mathrm{GGN}^{\mathrm{Data}}$ | mb | sub | $\mathrm{GGN}^{\mathrm{Data}}$ | mb | sub |
| exact | 13 | 87 | exact | 43 | 309 | exact | 35 | 135 | exact | 95 | 504 |
| mc | 640 | 959 | mc | 2015 | 2865 | mc | 1079 | 1536 | mc | 3360 | 3920 |

### B.2 Training of neural networks

**Procedure:** We train the following DEEPOBS (Schneider et al., 2019) architectures with SGD and ADAM: 3C3D on CIFAR-10, 2C2D on FASHION-MNIST and ALL-CNN-C on CIFAR-100 – all are equipped with cross-entropy loss. To ensure successful training, we use the hyperparameters from Dangel et al. (2020) (see Table S.3).

We also train a residual network RESNET-32 He et al. (2016) with cross-entropy loss on CIFAR-10 with both SGD and ADAM. For this, we use a batch size of 128 and train for 180 epochs. Momentum for SGD was fixed to 0.9, and ADAM uses the default parameters ($\beta_1 = 0.9$, $\beta_2 = 0.999$, $\epsilon = 10^{-8}$). For both optimizers, the learning rate was determined via grid search. Following (Schneider et al., 2019), we use a log-equidistant grid from $10^{-5}$ to $10^2$ and 36 grid points. As performance metric, the best test accuracy during training (evaluated once every epoch) is used.

**Results:** The results for the hyperparameter grid search are reported in Table S.3. The training metrics training/test loss/accuracy for all eight test problems are shown in Figure S.18 and S.19.

Table S.3: **Hyperparameter settings for training runs:** For both SGD and ADAM, we report their learning rates $\alpha$ (taken from the baselines in Dangel et al. (2020) or, for RESNET-32, determined via grid search). Momentum for SGD is fixed to 0.9. ADAM uses the default parameters $\beta_1 = 0.9$, $\beta_2 = 0.999$, $\epsilon = 10^{-8}$. We also report the batch size used for training and the number of training epochs.

| Problem | SGD | ADAM | Batch size | Train epochs |
|---|---|---|---|---|
| FASHION-MNIST 2C2D | $\alpha \approx 2.07 \cdot 10^{-2}$ | $\alpha \approx 1.27 \cdot 10^{-4}$ | $N = 128$ | 100 |
| CIFAR-10 3C3D | $\alpha \approx 3.79 \cdot 10^{-3}$ | $\alpha \approx 2.98 \cdot 10^{-4}$ | $N = 128$ | 100 |
| CIFAR-10 RESNET-32 | $\alpha \approx 6.31 \cdot 10^{-2}$ | $\alpha \approx 2.51 \cdot 10^{-3}$ | $N = 128$ | 180 |
| CIFAR-100 ALL-CNN-C | $\alpha \approx 4.83 \cdot 10^{-1}$ | $\alpha \approx 6.95 \cdot 10^{-4}$ | $N = 256$ | 350 |

### B.3 GGN vs. Hessian

**Checkpoints:** During training of the neural networks (see Appendix B.2), we store a copy of the model (i.e. the network's current parameters) at specific checkpoints. This grid defines the temporal resolution for all subsequent computations. Since training progresses much faster in the early training stages, we use a log-grid with 100 grid points between 1 and the number of training epochs and shift this grid by $-1$.

**Overlap:** Recall from Section 3.2: For the set of orthonormal eigenvectors $\{e_c^U\}_{c=1}^C$ to the $C$ largest eigenvalues of some symmetric matrix $U$, let $P^U = (e_1^U, ..., e_C^U)(e_1^U, ..., e_C^U)^\top$. As in Gur-Ari et al. (2018), the overlap between two subspaces $\mathcal{E}^U = \text{span}(e_1^U, ..., e_C^U)$ and $\mathcal{E}^V = \text{span}(e_1^V, ..., e_C^V)$ of the matrices $U$ and $V$ is defined by

$$\text{overlap}(\mathcal{E}^U, \mathcal{E}^V) = \frac{\text{Tr}(P^U P^V)}{\sqrt{\text{Tr}(P^U)\text{Tr}(P^V)}} \in [0, 1].$$

The overlap can be computed efficiently by using the trace's cyclic property: It holds $\text{Tr}(P^U P^V) = \text{Tr}(W^\top W)$ with $W = (e_1^U, ..., e_C^U)^\top(e_1^V, ..., e_C^V) \in \mathbb{R}^{C \times C}$. Note that this is a small $C \times C$ matrix, whereas $P^U, P^V \in \mathbb{R}^{D \times D}$. Since

$$\begin{aligned}
\text{Tr}(P^U) &= \text{Tr}((e_1^U, ..., e_C^U)(e_1^U, ..., e_C^U)^\top) \\
&= \text{Tr}((e_1^U, ..., e_C^U)^\top(e_1^U, ..., e_C^U)) && \text{(Cyclic property of trace)} \\
&= \text{Tr}(I_C) && \text{(Orthonormality of the eigenvectors)} \\
&= C
\end{aligned}$$

(and analogous $\text{Tr}(P^V) = C$), the denominator simplifies to $C$.

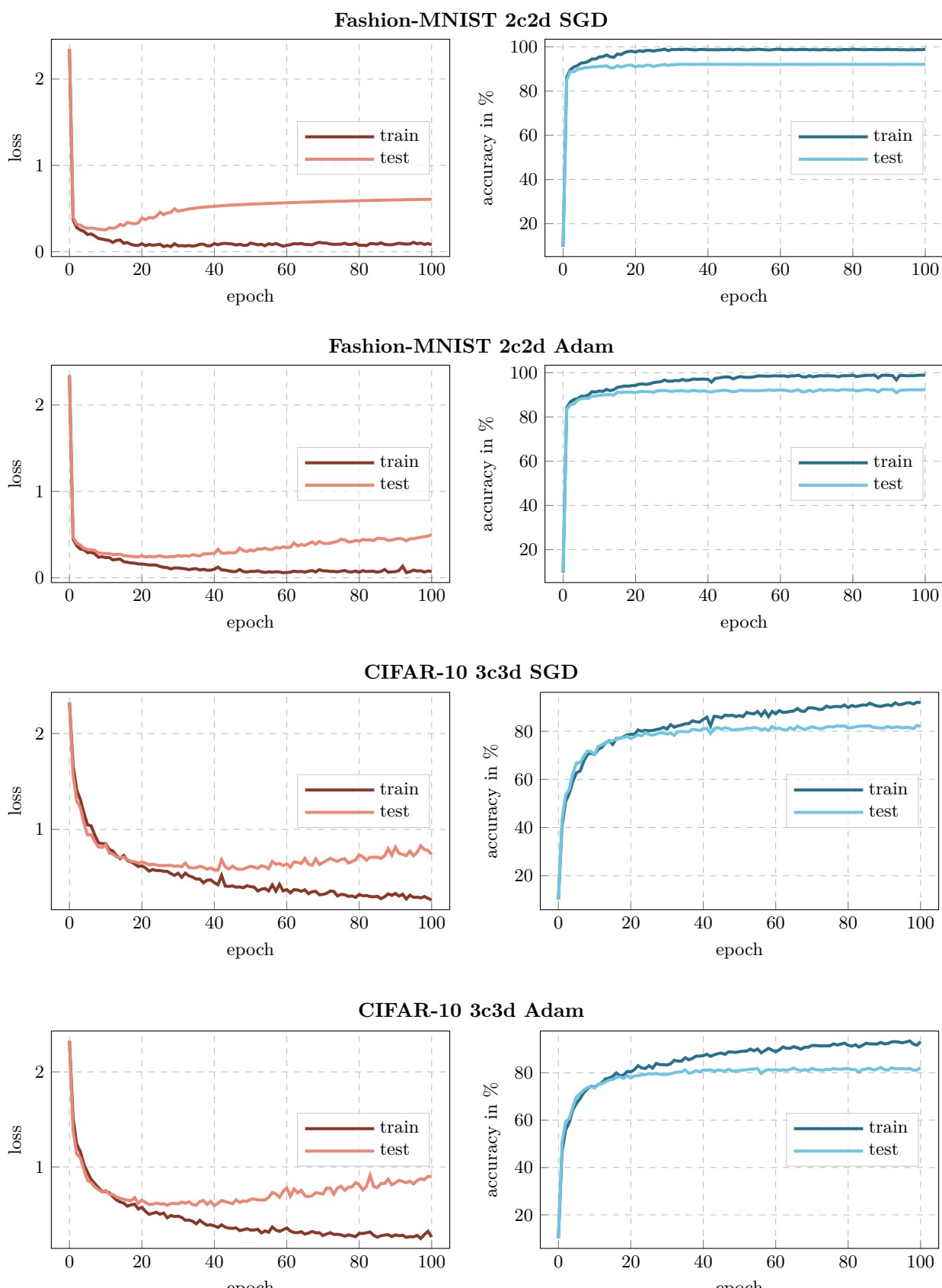

Figure S.18: **Training metrics (1):** Training/test loss/accuracy for all test problems.

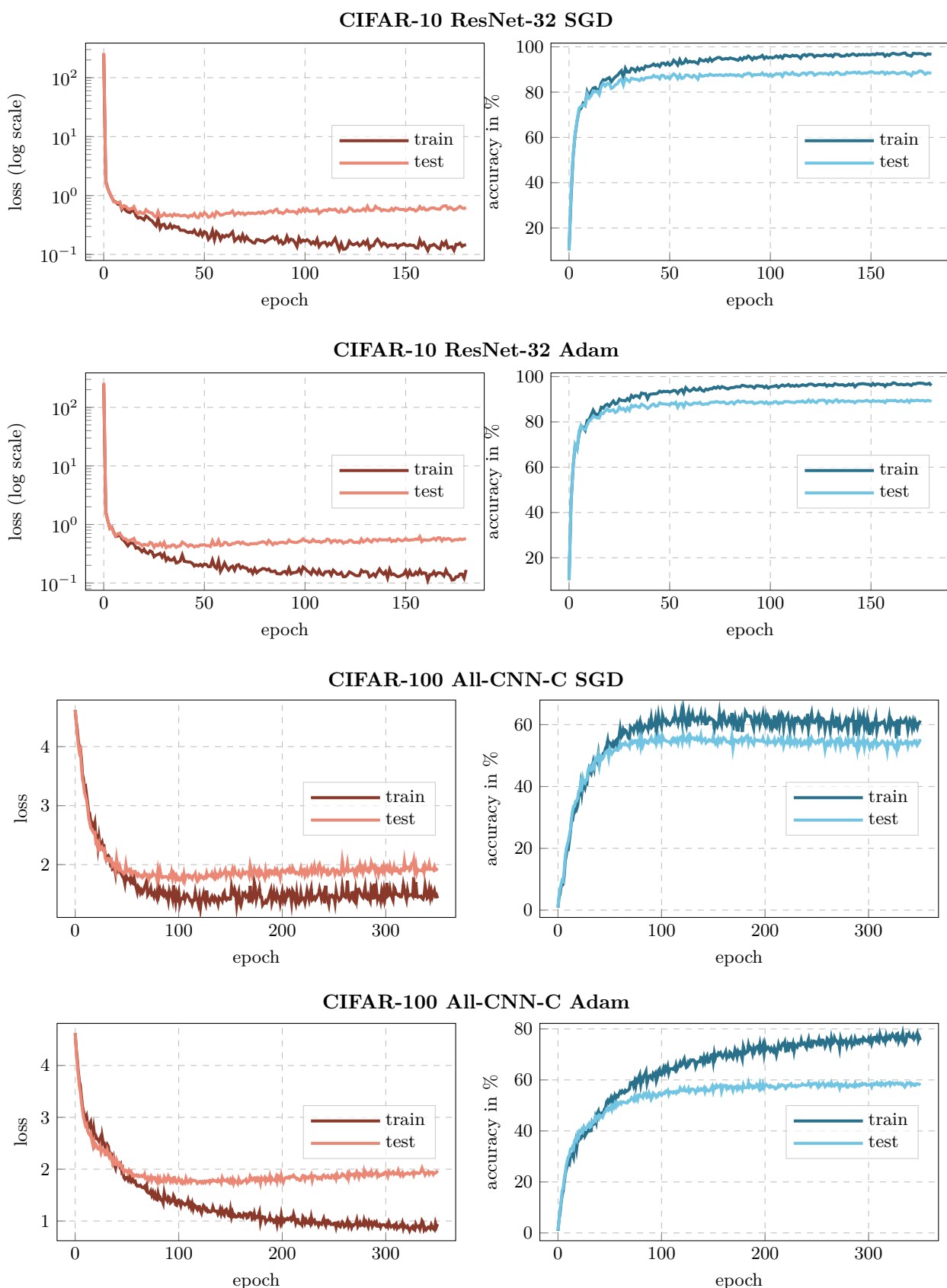

Figure S.19: **Training metrics (2):** Training/test loss/accuracy for all test problems.

**Procedure:** For each checkpoint, we compute the top-$C$ eigenvalues and associated eigenvectors of the full-batch GGN and Hessian (i.e. GGN and Hessian are both evaluated on the entire training set) using an iterative matrix-free approach. We then compute the overlap between the top-$C$ eigenspaces as described above. The eigspaces (i.e. the top-$C$ eigenvalues and associated eigenvectors) are stored on disk such that they can be used as a reference by subsequent experiments.

**Results:** The results for all test problems are presented in Figure S.20. Except for a short phase at the beginning of the optimization procedure (note the log scale for the epoch-axis), a strong agreement (note the different limits for the overlap-axis) between the top-$C$ eigenspaces is observed. We make similar observations for all test problems, yet to a slightly lesser extent for CIFAR-100. A possible explanation for this would be that the 100-dimensional eigenspaces differ in the eigenvectors associated with relatively small curvature. The corresponding eigenvalues already transition into the bulk of the spectrum, where the "sharpness of separation" decreases. However, since all directions are equally weighted in the overlap, overall slightly lower values are obtained.

### B.4 Eigenspace under noise and approximations

**Procedure (1):** We use the checkpoints and the definition of overlaps between eigenspaces from Appendix B.3. For the approximation of the GGN, we consider the cases listed in Table S.4.

Table S.4: **Considered cases for approximation of the eigenspace:** We use a different set of cases for the approximation of the GGN's full-batch eigenspace depending on the test problem. For the test problems with $C = 10$, we use $M = 1$ MC-sample, for the CIFAR-100 ALL-CNN-C test problem ($C = 100$), we use $M = 10$ MC-samples in order to reduce the computational costs by the same factor.

| Problem | Cases |
|---|---|
| FASHION-MNIST 2C2D CIFAR-10 3C3D and CIFAR-10 RESNET-32 | **mb, exact** with mini-batch sizes $N \in \{2, 8, 32, 128\}$ 
 **mb, mc** with $N = 128$ and $M = 1$ MC-sample 
 **sub, exact** using 16 samples from the mini-batch 
 **sub, mc** using 16 samples from the mini-batch and $M = 1$ MC-sample |
| CIFAR-100 ALL-CNN-C | **mb, exact** with mini-batch sizes $N \in \{2, 8, 32, 128\}$ 
 **mb, mc** with $N = 128$ and $M = 10$ MC-samples 
 **sub, exact** using 16 samples from the mini-batch 
 **sub, mc** using 16 samples from the mini-batch and $M = 10$ MC-samples |

For every checkpoint and case, we compute the top-$C$ eigenvectors of the respective approximation to the GGN. The eigenvectors are either computed directly using VIVIT (by transforming eigenvectors of the Gram matrix into parameter space, see Section 2.1) or, if not applicable (because memory requirements exceed $N_{\text{crit}}$, see Section 3.1), using an iterative matrix-free approach. The overlap is computed in reference to the GGN's full-batch top-$C$ eigenspace (see Appendix B.3). We extract 5 mini-batches from the training data and repeat the above procedure for each mini-batch (i.e. we obtain 5 overlap measurements for every checkpoint and case). The same 5 mini-batches are used over all checkpoints and cases.

**Results (1):** The results can be found in Figure S.21 and S.22. All test problems show the same characteristics: With decreasing computational effort, the approximation carries less and less structure of its full-batch counterpart, as indicated by dropping overlaps. In addition, for a fixed approximation method, a decrease in approximation quality can be observed over the course of training.

**Procedure (2):** Since VIVIT's GGN approximations using curvature sub-sampling and/or the MC approximation (the cases **mb, mc** as well as **sub, exact** and **sub, mc** in Table S.4) are based on the *mini*-batch GGN, we cannot expect them to perform better than this baseline. We thus repeat the analysis

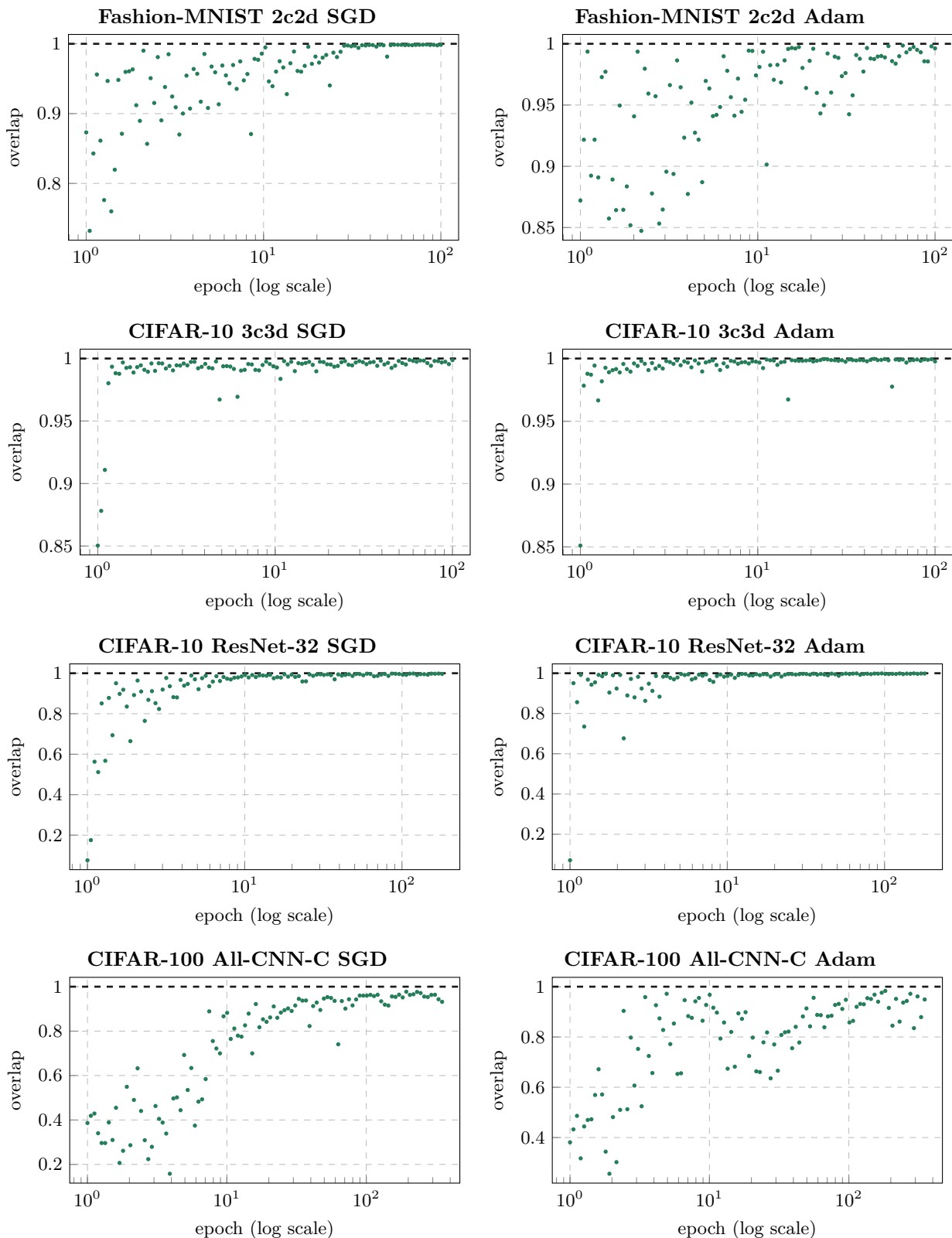

Figure S.20: **Full-batch GGN vs. full-batch Hessian:** Overlap between the top-$C$ eigenspaces of the full-batch GGN and full-batch Hessian during training for all test problems.

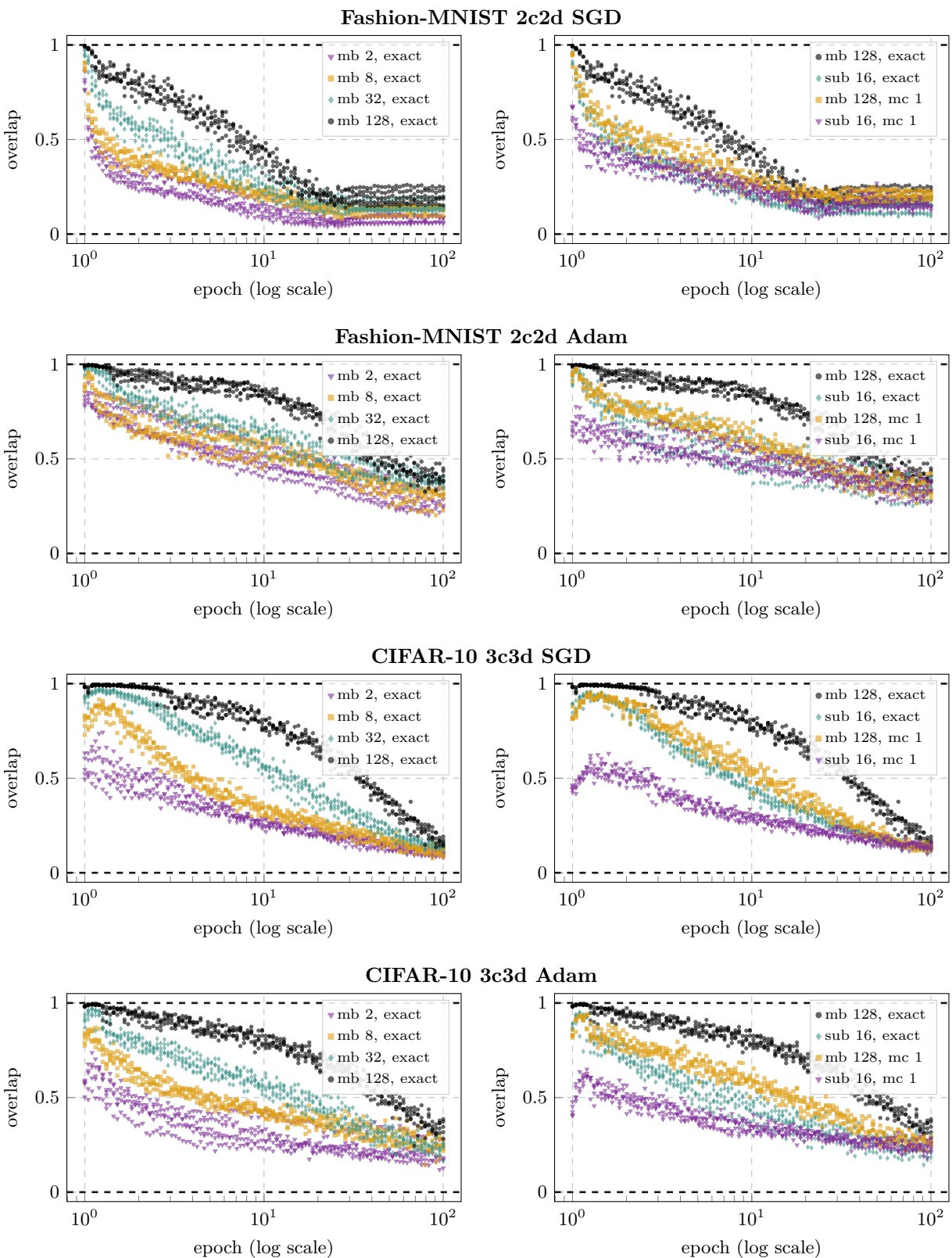

Figure S.21: **ViViT vs. full-batch GGN (1):** Overlap between the top-$C$ eigenspaces of different GGN approximations and the full-batch GGN during training for all test problems. Each approximation is evaluated on 5 different mini-batches.

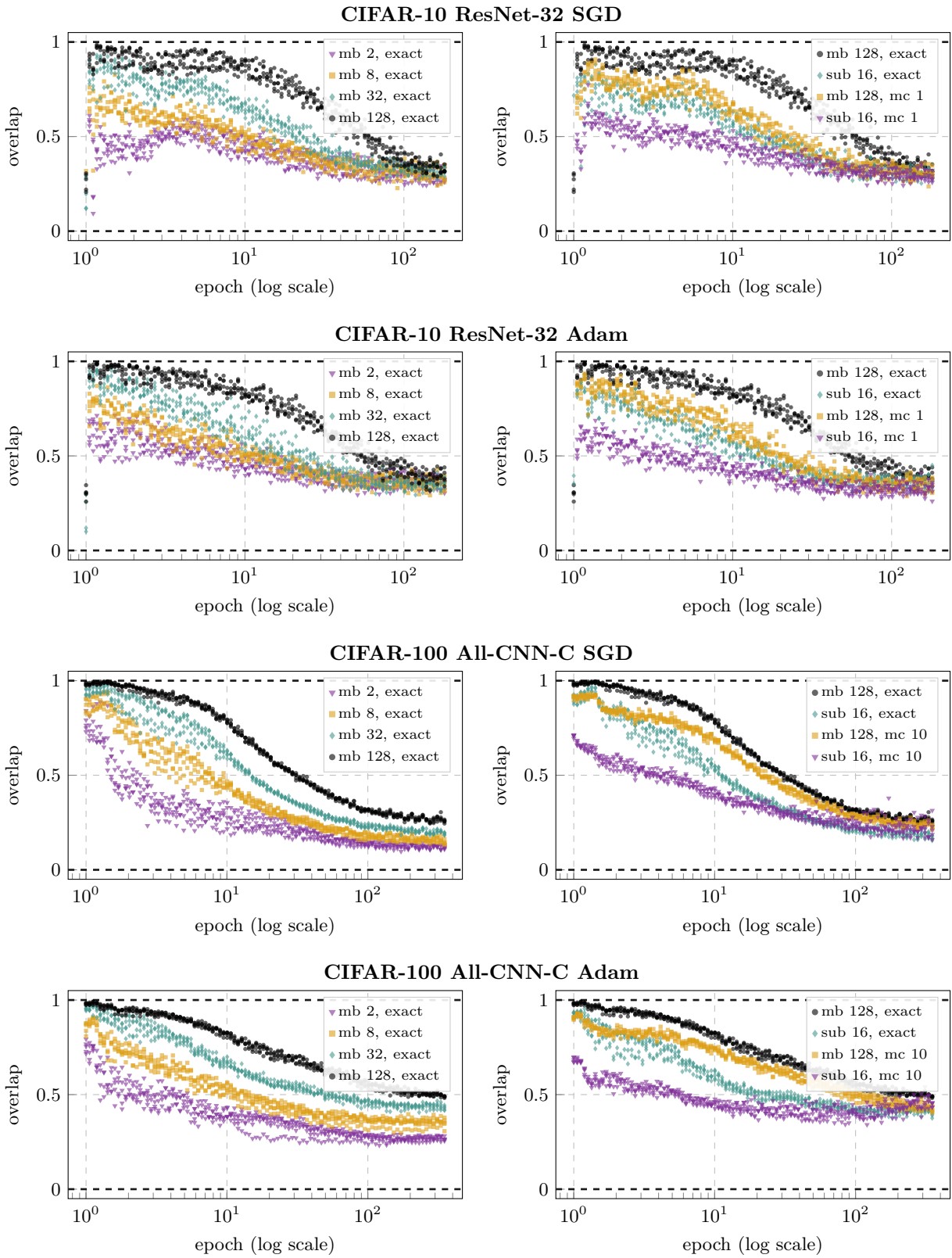

Figure S.22: **ViViT vs. full-batch GGN (2):** Overlap between the top-$C$ eigenspaces of different GGN approximations and the full-batch GGN during training for all test problems. Each approximation is evaluated on 5 different mini-batches.

from above but use the mini-batch GGN with batch-size $N = 128$ as ground truth instead of the full-batch GGN. Of course, the mini-batch reference top-$C$ eigenspace is always evaluated on the same mini-batch as the approximation.

**Results (2):**  The results can be found in Figure S.23. Over large parts of the optimization (note the log scale for the epoch-axis), the MC approximation seems to be better suited than curvature sub-sampling (which has comparable computational cost). For the CIFAR-100 ALL-CNN-C test problem, the MC approximation stands out particularly early from the other approximations and consistently yields higher overlaps with the mini-batch GGN.

### B.5    Curvature under noise and approximations

GGN and Hessian are predominantly used to locally approximate the loss by a quadratic model $q$ (see Equation (6)). Even if the curvature's eigenspace is completely preserved in spite of the approximations, they can still alter the curvature *magnitude* along the eigenvectors.

**Procedure:**  Table S.5 gives an overview over the cases considered in this experiment.

Table S.5: **Considered cases for approximation of curvature:** We use a different set of cases for the approximation of the GGN depending on the test problem. For the test problems with $C = 10$, we use $M = 1$ MC-sample, for the CIFAR-100 ALL-CNN-C test problem ($C = 100$), we use $M = 10$ MC-samples in order to reduce the computational costs by the same factor.

| Problem | Cases |
|---|---|
| FASHION-MNIST 2C2D CIFAR-10 3C3D and CIFAR-10 RESNET-32 | **mb, exact** with mini-batch size $N = 128$ 
 **mb, mc** with $N = 128$ and $M = 1$ MC-sample 
 **sub, exact** using 16 samples from the mini-batch 
 **sub, mc** using 16 samples from the mini-batch and $M = 1$ MC-sample |
| CIFAR-100 ALL-CNN-C | **mb, exact** with mini-batch size $N = 128$ 
 **mb, mc** with $N = 128$ and $M = 10$ MC-samples 
 **sub, exact** using 16 samples from the mini-batch 
 **sub, mc** using 16 samples from the mini-batch and $M = 10$ MC-samples |

Due to the large computational effort needed for the evaluation of full-batch directional derivatives, a subset of the checkpoints from Appendix B.3 is used for two test problems: We use every second checkpoint for CIFAR-10 RESNET-32 and every forth checkpoint for CIFAR-100 ALL-CNN-C.

For each checkpoint and case, we compute the top-$C$ eigenvectors $\{e_k\}_{k=1}^C$ of the GGN approximation $G^{(\mathrm{ap})}$ either with VIVIT or using an iterative matrix-free approach (as in Appendix B.4). The second-order directional derivative of the corresponding quadratic model along direction $e_k$ is then given by $\lambda_k^{(\mathrm{ap})} = e_k^\top G^{(\mathrm{ap})} e_k$ (see Equation (7)). As a reference, we compute the full-batch GGN $G^{(\mathrm{fb})}$ and the resulting directional derivatives along the same eigenvectors $\{e_k\}_{k=1}^C$, i.e. $\lambda_k^{(\mathrm{fb})} = e_k^\top G^{(\mathrm{fb})} e_k$. The average (over all $C$ directions) relative error is given by

$$\epsilon = \frac{1}{C} \sum_{k=1}^C \frac{|\lambda_k^{(\mathrm{ap})} - \lambda_k^{(\mathrm{fb})}|}{\lambda_k^{(\mathrm{fb})}} \, .$$

The procedure above is repeated on 3 mini-batches from the training data (i.e. we obtain 3 average relative errors for every checkpoint and case) – except for the CIFAR-100 ALL-CNN-C test problem, where we perform only a single run to keep the computational effort manageable.

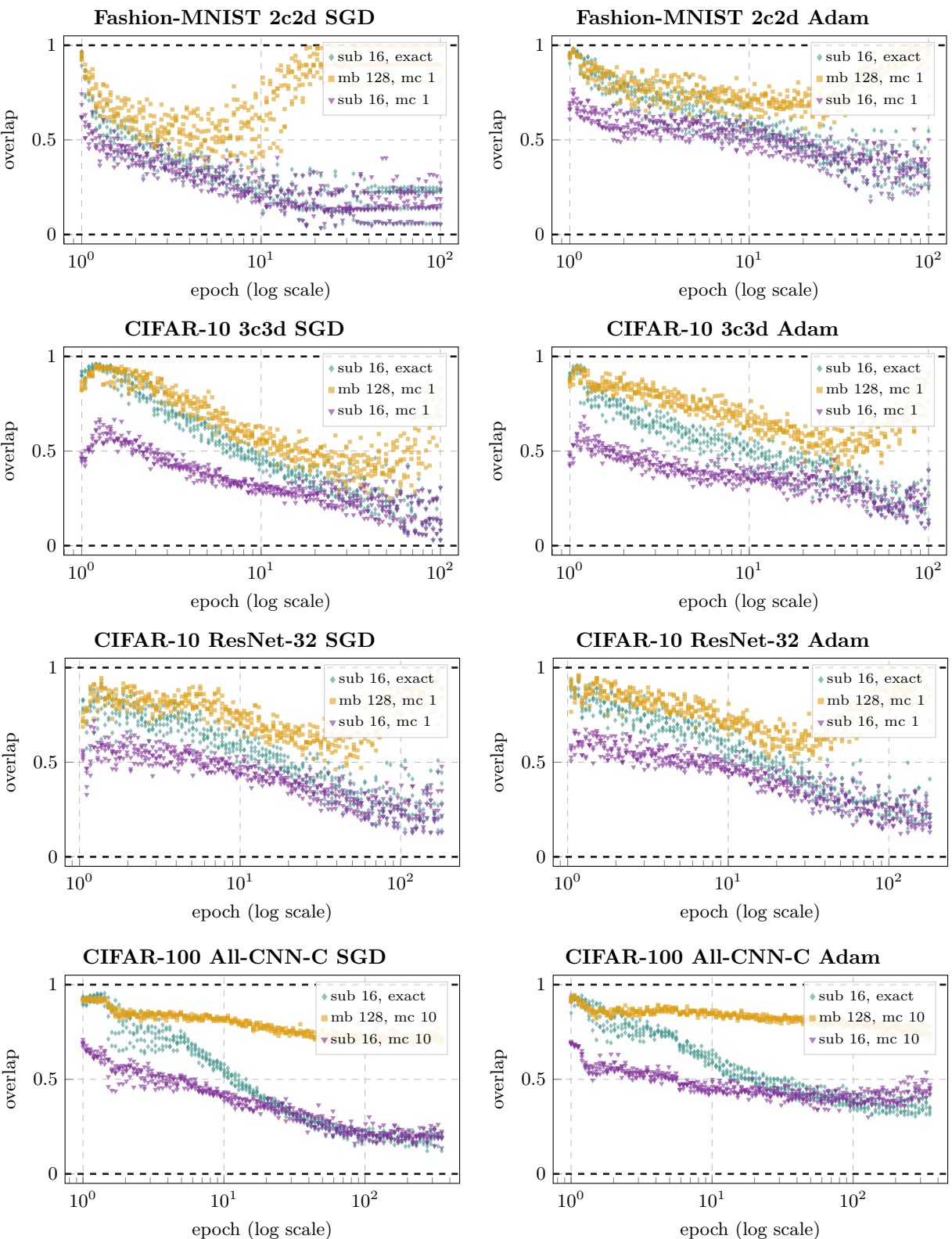

Figure S.23: **ViViT vs. mini-batch GGN:** Overlap between the top-$C$ eigenspaces of different GGN approximations and the mini-batch GGN during training for all test problems. Each approximation is evaluated on 5 different mini-batches.

**Results:** The results can be found in Figure S.24. We observe similar results as in Appendix B.4: With increasing computational effort, the approximated directional derivatives become more precise and the overall approximation quality decreases over the course of the optimization. For the RESNET-32 architecture, the average errors are particularly large.

### B.6 Directional derivatives

**Procedure:** We use the checkpoints from Appendix B.3. For every checkpoint, we compute the top-$C$ eigenvectors of the mini-batch GGN (with a mini-batch size of $N = 128$) using an iterative matrix-free method. We also compute the mini-batch gradient. The first- and second-order directional derivatives of the resulting quadratic model (see Equation (6)) are given by Equation (8).

We use these directional derivatives $\{\gamma_{nk}\}_{n=1,k=1}^{N,C}$, $\{\lambda_{nk}\}_{n=1,k=1}^{N,C}$ to compute signal-to-noise ratios (SNRs) along the top-$C$ eigenvectors. The curvature SNR along direction $\boldsymbol{e}_k$ is given by the squared sample mean divided by the empirical variance of the samples $\{\lambda_{nk}\}_{n=1}^{N}$, i.e.

$$\text{SNR} = \frac{\lambda_k^2}{1/N-1 \sum_{n=1}^{N}(\lambda_{nk} - \lambda_k)^2} \quad \text{where} \quad \lambda_k = \frac{1}{N}\sum_{n=1}^{N}\lambda_{nk}\,.$$

(and similarly for $\{\gamma_{nk}\}_{n=1}^{N}$).

**Results:** The results can be found in Figure S.25 and S.26. These plots show the SNRs in $C$ distinct colors that are generated by linearly interpolating in the RGB color space from black (●) to light red (●). At each checkpoint, the colors are assigned based on the *order* of the respective directional curvature $\lambda_k$: The SNR that belongs to the direction with the smallest curvature is shown in black and the SNR that belongs to the direction with the largest curvature is shown in light red. The color thus encodes only the order of the top-$C$ directional curvatures – *not* their magnitude. We use this color encoding to reveal potential correlations between SNR and curvature.

We find that the gradient SNR along the top-$C$ eigenvectors is consistently small (in comparison to the curvature SNR) and remains roughly on the same level during the optimization. The curvature signal decreases as training proceeds. The SNRs along the top-C eigendirections do not appear to show any significant correlation with the corresponding curvatures. Only for the CIFAR-100 test problems we can suspect a correlation between strong curvature and small curvature SNR.

## C Implementation details

**Layer view of backpropagation:** Consider a single layer $T_{\boldsymbol{\theta}^{(i)}}^{(i)}$ that transforms inputs $\boldsymbol{z}_n^{(i-1)} \in \mathbb{R}^{h^{(i-1)}}$ into outputs $\boldsymbol{z}_n^{(i)} \in \mathbb{R}^{h^{(i)}}$ by means of a parameter $\boldsymbol{\theta}^{(i)} \in \mathbb{R}^{d^{(i)}}$. During backpropagation for $\boldsymbol{V}$, the layer receives vectors $\boldsymbol{s}_{nc}^{(i)} = (\mathrm{J}_{\boldsymbol{z}_n^{(i)}} f_n)^\top \boldsymbol{s}_{nc}$ from the previous stage (recall $\nabla_f^2 \ell_n = \sum_{c=1}^{C} \boldsymbol{s}_{nc}\boldsymbol{s}_{nc}^\top$). Parameter contributions $\boldsymbol{v}_{nc}^{(i)}$ to $\boldsymbol{V}$ are obtained by application of its Jacobian,

$$\begin{aligned}
\boldsymbol{v}_{nc}^{(i)} &= (\mathrm{J}_{\boldsymbol{\theta}^{(i)}} f_n)^\top \boldsymbol{s}_{nc} \\
&= \left(\mathrm{J}_{\boldsymbol{\theta}^{(i)}} \boldsymbol{z}_n^{(i)}\right)^\top \left(\mathrm{J}_{\boldsymbol{z}_n^{(i)}} f_n\right)^\top \boldsymbol{s}_{nc} && \text{(Chain rule)} \\
&= \left(\mathrm{J}_{\boldsymbol{\theta}^{(i)}} \boldsymbol{z}_n^{(i)}\right)^\top \boldsymbol{s}_{nc}^{(i)}\,. && \text{(Definition of } \boldsymbol{s}_{nc}^{(i)}\text{)}
\end{aligned} \qquad (\text{S}.11)$$

Consequently, the contribution of $\boldsymbol{\theta}^{(i)}$ to $\boldsymbol{V}$, denoted by $\boldsymbol{V}^{(i)} \in \mathbb{R}^{d^{(i)} \times NC}$, is

$$\boldsymbol{V}^{(i)} = \frac{1}{\sqrt{N}} \begin{pmatrix} \boldsymbol{v}_{11}^{(i)} & \boldsymbol{v}_{12}^{(i)} & \dots & \boldsymbol{v}_{NC}^{(i)} \end{pmatrix} \quad \text{with} \quad \boldsymbol{v}_{nc}^{(i)} = (\mathrm{J}_{\boldsymbol{\theta}^{(i)}} f_n)^\top \boldsymbol{s}_{nc}\,. \qquad (\text{S}.12)$$

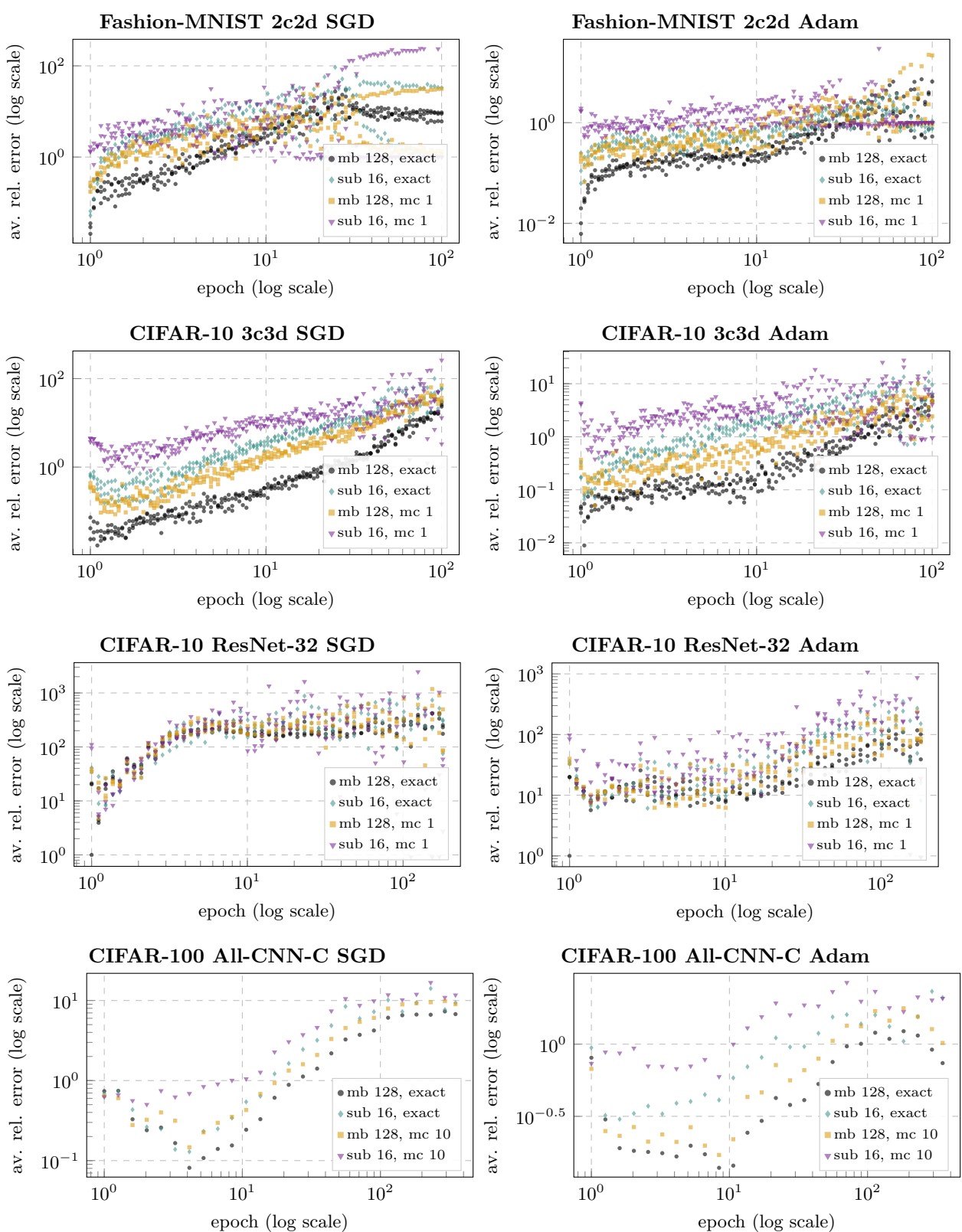

Figure S.24: **ViViT's vs. full-batch quadratic model:** Comparison between approximations to the quadratic model and the full-batch model in terms of the average relative error for the directional curvature during training for all test problems.

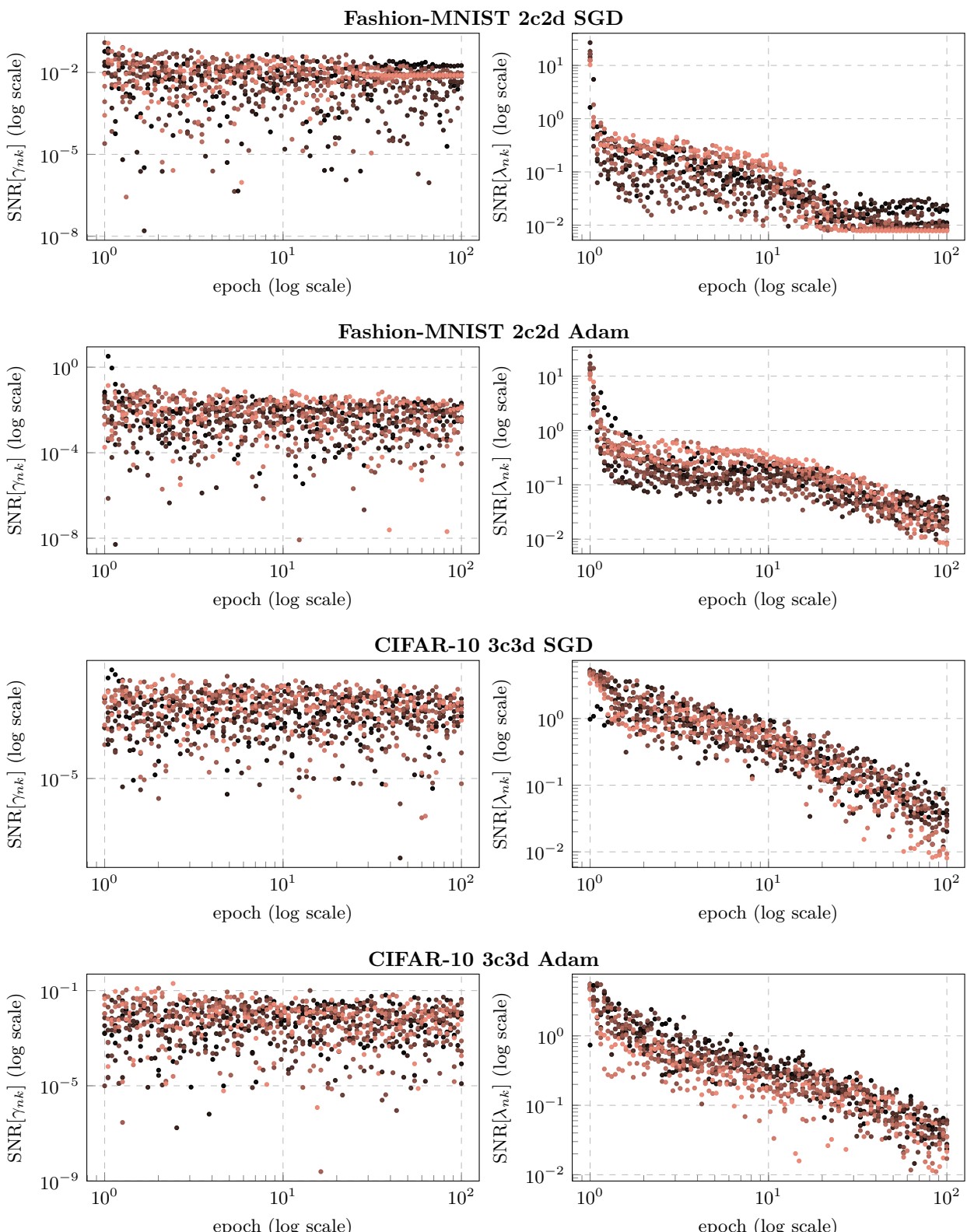

Figure S.25: **Directional SNRs (1):** SNR along each of the mini-batch GGN's top-$C$ eigenvectors during training for all test problems. At fixed epoch, the SNR for the most curved direction is shown in ● and for the least curved direction in ●.

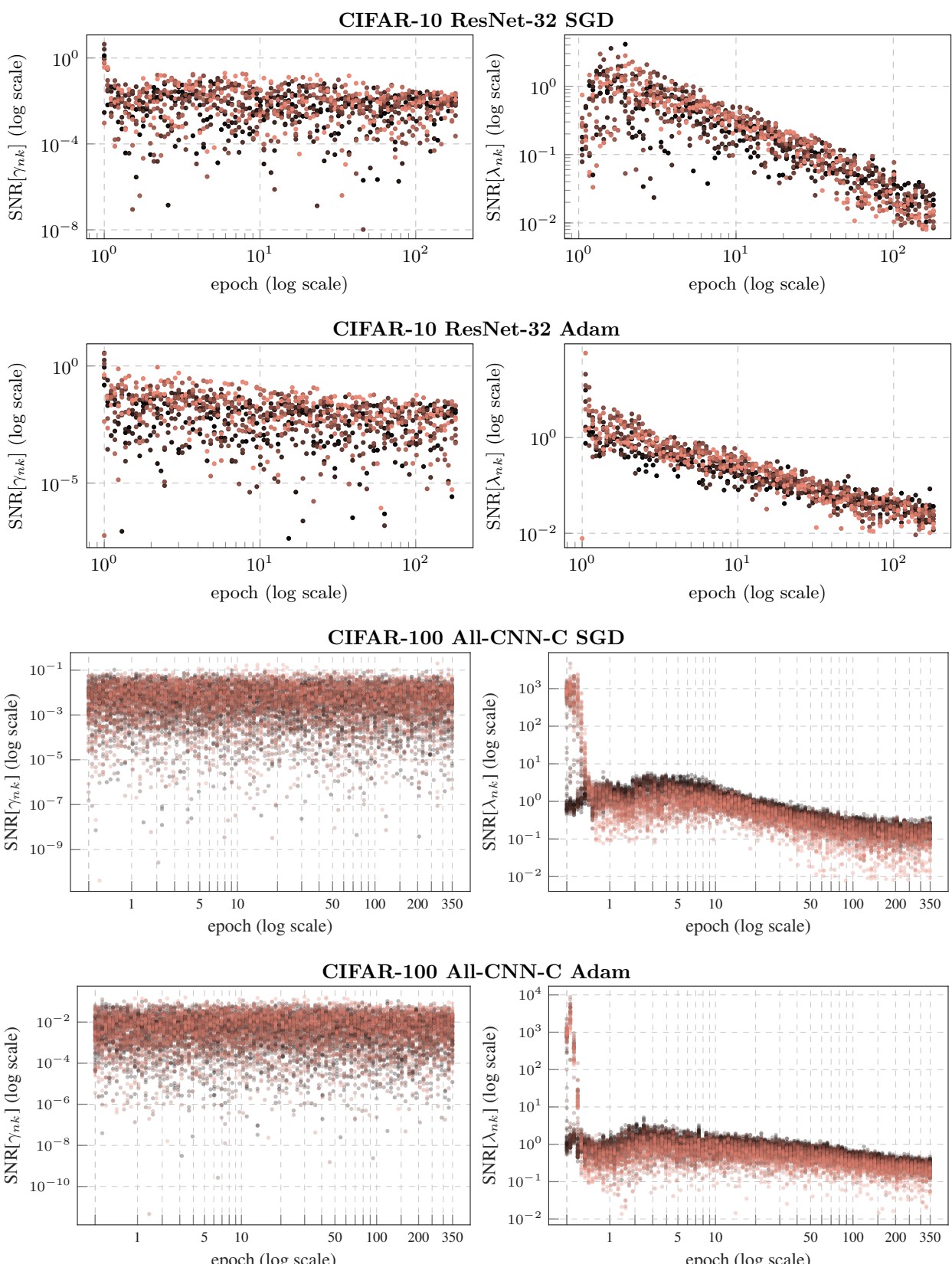

Figure S.26: **Directional SNRs (2):** SNR along each of the mini-batch GGN's top-$C$ eigenvectors during training for all test problems. At fixed epoch, the SNR for the most curved direction is shown in ● and for the least curved direction in ●.

### C.1 Optimized Gram matrix computation for linear layers

Our goal is to efficiently extract $\boldsymbol{\theta}^{(i)}$'s contribution to the Gram matrix $\tilde{\mathbf{G}}$, given by

$$\tilde{\mathbf{G}}^{(i)} = \boldsymbol{V}^{(i)\top} \boldsymbol{V}^{(i)} \in \mathbb{R}^{NC \times NC} . \tag{S.13}$$

**Gram matrix via expanding $\boldsymbol{V}^{(i)}$:** One way to construct $\boldsymbol{G}^{(i)}$ is to first expand $\boldsymbol{V}^{(i)}$ (Equation (S.12)) via the Jacobian $\mathrm{J}_{\boldsymbol{\theta}^{(i)}} \boldsymbol{z}_n^{(i)}$, then contract it (Equation (S.13)). This can be a memory bottleneck for large linear layers which are common in many architectures close to the network output. However if only the Gram matrix rather than $\boldsymbol{V}$ is required, structure in the Jacobian can be used to construct $\tilde{\mathbf{G}}^{(i)}$ without expanding $\boldsymbol{V}^{(i)}$ and thus reduce this overhead.

**Optimization for linear layers:** Now, let $T_{\boldsymbol{\theta}^{(i)}}^{(i)}$ be a linear layer with weights $\boldsymbol{W}^{(i)} \in \mathbb{R}^{h^{(i)} \times h^{(i-1)}}$, i.e. $\boldsymbol{\theta}^{(i)} = \mathrm{vec}(\boldsymbol{W}^{(i)}) \in \mathbb{R}^{d^{(i)} = h^{(i)} h^{(i-1)}}$ with column stacking convention for vectorization,

$$T_{\boldsymbol{\theta}^{(i)}}^{(i)} : \quad \boldsymbol{z}_n^{(i)} = \boldsymbol{W}^{(i)} \boldsymbol{z}_n^{(i-1)} .$$

The Jacobian is

$$\mathrm{J}_{\boldsymbol{\theta}^{(i)}} \boldsymbol{z}_n^{(i)} = \boldsymbol{z}_n^{(i-1)\top} \otimes \boldsymbol{I}_{h^{(i)}} . \tag{S.14}$$

Its structure can be used to directly compute entries of the Gram matrix without expanding $\boldsymbol{V}^{(i)}$,

$$
\begin{aligned}
\left[\tilde{\mathbf{G}}^{(i)}\right]_{(nc)(n'c')} &= \boldsymbol{v}_{nc}^{(i)\top} \boldsymbol{v}_{n'c'}^{(i)} && \text{(Equation (S.13))}\\
&= \boldsymbol{s}_{nc}^{(i)\top} \left(\mathrm{J}_{\boldsymbol{\theta}^{(i)}} \boldsymbol{z}_n^{(i)}\right) \left(\mathrm{J}_{\boldsymbol{\theta}^{(i)}} \boldsymbol{z}_{n'}^{(i)}\right)^\top \boldsymbol{s}_{n'c'}^{(i)} \\
&= \boldsymbol{s}_{nc}^{(i)\top} \left(\boldsymbol{z}_n^{(i-1)\top} \otimes \boldsymbol{I}_{h^{(i)}}\right) \left(\boldsymbol{z}_{n'}^{(i-1)\top} \otimes \boldsymbol{I}_{h^{(i)}}\right)^\top \boldsymbol{s}_{n'c'}^{(i)} && \text{(Equation (S.14))} \\
&= \boldsymbol{s}_{nc}^{(i)\top} \left(\boldsymbol{z}_n^{(i-1)\top} \boldsymbol{z}_{n'}^{(i-1)} \otimes \boldsymbol{I}_{h^{(i)}}\right) \boldsymbol{s}_{n'c'}^{(i)} && \text{(Equation (S.11))} \\
&= \boldsymbol{z}_n^{(i-1)\top} \boldsymbol{z}_{n'}^{(i-1)} \boldsymbol{s}_{nc}^{(i)\top} \boldsymbol{I}_{h^{(i)}} \boldsymbol{s}_{n'c'}^{(i)} && (\boldsymbol{z}_n^{(i-1)\top} \boldsymbol{z}_{n'}^{(i-1)} \in \mathbb{R}) \\
&= \left(\boldsymbol{z}_n^{(i-1)\top} \boldsymbol{z}_{n'}^{(i-1)}\right) \left(\boldsymbol{s}_{nc}^{(i)\top} \boldsymbol{s}_{n'c'}^{(i)}\right) .
\end{aligned}
$$

We see that the Gram matrix is built from two Gram matrices based on $\{\boldsymbol{z}_n^{(i-1)}\}_{n=1}^N$ and $\{\boldsymbol{s}_{nc}^{(i)}\}_{n=1,c=1}^{N,C}$, that require $\mathcal{O}(N^2)$ and $\mathcal{O}((NC)^2)$ memory, respectively. In comparison, the naïve approach via $\boldsymbol{V}^{(i)} \in \mathbb{R}^{d^{(i)} \times NC}$ scales with the number of weights, which is often comparable to $D$. For instance, the 3C3D architecture on CIFAR-10 has $D = 895{,}210$ and the largest weight matrix has $d^{(i)} = 589{,}824$, whereas $NC = 1{,}280$ during training (Schneider et al., 2019).

**Run time comparison:** To evaluate the efficiency of our optimization for linear layers, we consider a multi-layer perceptron with $1024 \to 512 \to 256 \to 128 \to 64 \to 32 \to C = 10$ units, each activated by ReLU except for the last, and mean squared error as loss function. We set the batch size to $N = 128$ and randomly generate synthetic inputs and labels. The following tasks are considered (results are for GPU, reporting the smallest run time over 20 repetitions):

- T1: Mini-batch gradient computation: $t_{\mathrm{grad}} \approx 0.774\,\mathrm{ms}$ (1.0 x)

- T2: Naive computation of *explicit* $\boldsymbol{V}$: $t_{\boldsymbol{V}} \approx 9.25\,\mathrm{ms}$ (11.9 x)

- T3: Naive computation of the Gram matrix $\tilde{\mathbf{G}}$ via *explicit* $\boldsymbol{V}$: $t_{\boldsymbol{V},\tilde{\mathbf{G}}} \approx 233\,\mathrm{ms}$ (301 x)

- T4: Optimized computation of $\tilde{\mathbf{G}}$ via *implicit* $\boldsymbol{V}$: $t_{\boldsymbol{V},\tilde{\mathbf{G}},\mathrm{opt}} \approx 4.28\,\mathrm{ms}$ (5.5 x)

The main finding of this comparison is that, for this setting, **our optimized approach (T4) computes the Gram matrix >50 x faster than the naive approach (T3), and requires less than $C$ gradient computations (T1)**. Detailed description:

- **Naive computations of $V$ (T2) and $\tilde{\mathbf{G}}$ (T3) are slow:** From the complexity analysis in Section 2.3, we expect $t_V \approx C t_{\text{grad}}$. Empirically, we observe a factor of 11.9. Despite the use of a GPU, performance does not improve. We believe this may be because the operation requires allocating and writing large amounts of data into memory for $V$. Naively expanding this *explicit* $V$, then computing $\tilde{\mathbf{G}}$ (T3), would cost $t_{V,G}/t_{\text{grad}} \approx 301$ gradients.

- **Our optimized approach to *implicitly* work with $V$ in combination with exploiting structure in the Jacobian (T4) is fast:** With our optimizations, we observe $t_{V,\tilde{\mathbf{G}},\text{opt}}/t_{\text{grad}} \approx 5.5$. This is a $> 50$ x speed-up over the naive approach (T3)! It also requires less than $C$ gradient computations, i.e. almost 50% of the expected overhead is compensated by parallelism on the GPU.

In our work, we first extended the BACKPACK package by the naive operations T2 and T3 described above. These are already superior to for-loop based approaches with PYTORCH's built-in automatic differentiation because BACKPACK implements vectorized vector-Jacobian products. On top of that, we then further *significantly* improved performance through structural tricks (T4), as shown above.

### C.2 Implicit multiplication with the inverse (block-diagonal) GGN

**Inverse GGN-vector products:** A damped Newton step requires multiplication by $(G + \delta I_D)^{-1}$.[8] By means of Equation (3) and the matrix inversion lemma,

$$
\begin{aligned}
(\delta I_D + G)^{-1} &= \left(\delta I_D + V V^\top\right)^{-1} &&\text{(Equation (3))}\\
&= \frac{1}{\delta}\left(I_D + \frac{1}{\delta} V V^\top\right)^{-1}\\
&= \frac{1}{\delta}\left[I_D - \frac{1}{\delta} V \left(I_{NC} + V^\top \frac{1}{\delta} V\right)^{-1} V^\top\right] &&\text{(Matrix inversion lemma)}\\
&= \frac{1}{\delta}\left[I_D - V \left(\delta I_{NC} + V^\top V\right)^{-1} V^\top\right] &&\text{(Gram matrix)}\\
&= \frac{1}{\delta}\left[I_D - V \left(\delta I_{NC} + \tilde{\mathbf{G}}\right)^{-1} V^\top\right].
\end{aligned}
$$
(S.15)

Inverse GGN-vector products require inversion of the damped Gram matrix as well as applications of $V, V^\top$ for the transformations between Gram and parameter space.

**Inverse block-diagonal GGN-vector products:** Next, we replace the full GGN by its block diagonal approximation $G \approx G_{\text{BDA}} = \text{diag}(G^{(1)}, G^{(2)}, \dots)$ with

$$
G^{(i)} = V^{(i)} V^{(i)^\top} \in \mathbb{R}^{d^{(i)} \times d^{(i)}}
$$

and $V^{(i)}$ as in Equation (S.12). Then, inverse multiplication reduces to each block,

$$
G_{\text{BDA}}^{-1} = \text{diag}\left(G^{(1)^{-1}}, G^{(2)^{-1}}, \dots\right).
$$

If again a damped Newton step is considered, we can reuse Equation (S.15) with the substitutions

$$
\left(G, D, V, V^\top, \tilde{\mathbf{G}}\right) \leftrightarrow \left(G^{(i)}, d^{(i)}, V^{(i)}, V^{(i)^\top}, \tilde{\mathbf{G}}^{(i)}\right)
$$

to apply the inverse and immediately discard the VIVIT factors: At backpropagation of layer $T_{\boldsymbol{\theta}^{(i)}}^{(i)}$

---

[8] $\delta I_D$ can be replaced by other easy-to-invert matrices.

1. Compute $\boldsymbol{V}^{(i)}$ using Equation (S.12).

2. Compute $\tilde{\mathbf{G}}^{(i)}$ using Equation (S.13).

3. Compute $\left(\delta \boldsymbol{I}_{NC} + \tilde{\mathbf{G}}^{(i)}\right)^{-1}$.

4. Apply the inverse in Equation (S.15) with the above substitutions to the target vector.

5. Discard $\boldsymbol{V}^{(i)}$, $\boldsymbol{V}^{(i)\top}$, $\tilde{\mathbf{G}}^{(i)}$, and $\left(\delta \boldsymbol{I}_{NC} + \tilde{\mathbf{G}}^{(i)}\right)^{-1}$. Proceed to layer $i - 1$.

Note that the above scheme should only be used for parameters that satisfy $d^{(i)} > NC$, i.e. $\dim(\boldsymbol{G}^{(i)}) > \dim(\tilde{\mathbf{G}}^{(i)})$. Low-dimensional parameters can be grouped with others to increase their joint dimension, and to control the block structure of $\boldsymbol{G}_{\mathrm{BDA}}$.

