# OpenReview forum: "ViViT: Curvature Access Through The Generalized Gauss-Newton’s Low-Rank Structure"
_TMLR — Accepted by TMLR_

### Review · Reviewer_kVKm · 2022-11-12

**Summary Of Contributions:**

This paper proposed a method ViViT to compute the eigenvalues and eigenvectors of the generalized Gauss-Newton (GGN) approximation of neural network Hessian matrices. ViViT is more efficient than previous methods by utilizing the low-rank nature of GGN, and it has approximation versions that are more efficient than ViViT but are less accurate. The authors also did experiments for convolutional networks on FASHION-MNIST and CIFAR to verify the efficiency of ViViT and analyze the factors that influence the performance of ViViT.

**Audience:**

Yes

**Broader Impact Concerns:**

This paper focuses on a general problem of computing and approximating the second-order information of neural network training, and I do not see any immediate concerns about the ethical implications of the work.

**Claims And Evidence:**

Yes

**Requested Changes:**

- It could be better if the authors could add theoretical or empirical justifications for the approximated versions of ViViT.

- It might be better to provide the running time comparison to gradient computation.

- Figure 1b seems a bit confusing. It looks like an overlap of two figures, and the curves for $\gamma_k$ and $\lambda_k$ are partially hidden. Could the authors add more explanation on the figure or in the text description?

**Strengths And Weaknesses:**

Strengths:

- The efficiency of ViViT is better than previous methods in computing top-$k$ eigenvectors when $k$ is larger than 1, and the computation time almost doesn't grow with $k$. This makes ViViT useful for computing the top eigenspaces of neural network Hessian.

- This paper is clearly written. The problem background is introduced, the notations and settings are well-defined, and the sections are well-organized and form a smooth logic flow. The related works appear to be properly cited.

- The authors empirically studied not only the efficiency and approximation accuracy of ViViT, but also the factors influencing its performance, including the variance in first and second-order directional derivatives during training. This could help with the understanding of neural network Hessians.

Weaknesses:

- There might not be enough theoretical or empirical guarantees for the performance of the approximated versions of ViViT. The accuracy of the approximation depends on the distribution of the vectors $s_{nc}$, which is related to the data distribution and network structure and weights. The experiments measure the performance by top-$C$ eigenspace overlap on standard convolutional networks, and it could be hard to justify whether it can be generalized to more network structures or beyond top-$C$ eigenspaces.

- The wallclock running time of ViViT compared to that of backpropagation is not provided. Although ViViT can utilize the intermediate results from the computation of gradients, it might be unclear how much overhead it introduces in practice.

---

> ### Author Response · Authors · 2022-11-17
> **Rebuttal**
>
> Dear Reviewer kVKm,
>
> Thanks a lot for your constructive feedback. We updated our manuscript to
> include an improved description of Figure 1b in the caption. Please let us know
> if you are satisfied with this change.
>
> In the following, we would also like to address your remaining points:
>
> - *"There might not be enough theoretical or empirical guarantees for the
> performance of the approximated versions of ViViT."*
>
>     Theoretically (under the assumption of i.i.d. samples), both approximations
>     to reduce computation (sub-sampling and MC-sampling) yield unbiased Monte
>     Carlo estimators for the GGN. We tried to motivate those approaches through
>     existing works:
>
>     (i) For example, curvature sub-sampling is a common technique in both
>     theoretical [1] and practical [2] works on second-order methods.
>
>     (ii) For the loss functions studied in our work, the MC approximation of the
>     GGN corresponds to a principled scheme for approximating the Fisher
>     information matrix by sampling from the model's likelihood implied by the
>     loss. This scheme is used by seminal works on approximating the Fisher/GGN
>     [3, 4]. The MC-approximated quantities have been found to work well
>     empirically in the context of second-order optimization [5].
>
> - *"[I]t could be hard to justify whether it [the experiment] can be
> generalized to more network structures or beyond top-$C$ eigenspaces"*
>
>     We understand this concern. Our experiments try to achieve a large coverage
>     of representative tasks (different image classification problems),
>     architectures (convolutional neural networks, ResNets), and training
>     algorithms (SGD, Adam) to obtain representative results.
>
>     Besides practical computational concerns, our decision to limit the
>     investigation of approximation quality to the top-$C$ eigenspace is
>     motivated by the findings of [6]. This work empirically shows that the
>     gradient, and therefore contemporary training, resides mostly in this space.
>     Hence, interactions between gradient and curvature in exactly this subspace
>     are of particular interest for studying phenomena like implicit
>     regularisation, or gradient preconditioning with curvature information.
>
> - *"The wall clock running time of ViViT compared to that of backpropagation is
> not provided."*
>
>     We understand your point that, in the main text, we do not show (or rescale
>     the run time by) the time to compute a mini-batch gradient. Our runtime
>     comparison (e.g. Figure 2) aims at highlighting the comparison with a
>     competitor method that seeks to compute the same quantity of interest, i.e.
>     curvature information. Note that re-scaling the time axis by the gradient
>     computation time would not alter the results of this comparison.
>
>     However, our manuscript includes a concrete evaluation of the practical
>     overhead introduced by ViViT: in Appendix C.1, we present an in-depth
>     comparison of ViViT and gradient backpropagation, and evaluate the overhead
>     ViViT introduces in practice. We empirically observe that ViViT's run time overhead is smaller than
>     expected (based on the added computational work):
>
>     For the investigated setting ($C=10$), we would expect that computing
>     $\mathbf{V}$ costs $C=10$ gradient computations. In practice, we observe
>     that we can form the Gram matrix $\mathbf{V}^\top \mathbf{V}$ (which
>     implicitly involves forming **and** taking the inner product of
>     $\mathbf{V}$) in $\approx 5.5$ gradient computations, which corresponds to a
>     reduction in overhead of $\approx 50\\%$. This speed-up is attributable to
>     recycling, parallelism, and our optimized implementation.
>
> We would be happy to further discuss.
>
> ---
>
> [1] Roosta-Khorasani, F., & Mahoney, M. W. (2019). Sub-sampled newton methods.
>
> [2] Zhang, H., Xiong, C., Bradbury, J., & Socher, R. (2017). Block-diagonal
>   Hessian-free optimization for training neural networks.
>
> [3] Martens, J., & Grosse, R. (2015). Optimizing neural networks with
>   Kronecker-factored approximate curvature.
>
> [4] Grosse, R., & Martens, J. (2016). A kronecker-factored approximate Fisher
>   matrix for convolution layers.
>
> [5] Dangel, F., Kunstner, F., & Hennig, P. (2020). BackPACK: packing more into
>   backprop.
>
> [6] Gur-Ari, G., Roberts, D. A., & Dyer, E. (2018). Gradient descent happens in a
>   tiny subspace.

---

> > ### Comment · Reviewer_kVKm · 2022-11-26
> > **Update after author response**
> >
> > Thank the authors for the very detailed response. I have read the updated draft, all other reviews, and the authors' responses. My original major concerns are mostly addressed by the authors' responses.
> >
> > Figure 1b looks clear to me after the caption revision, and the motivations from existing works provide some justification for the sub-sampling methods used in this paper. It is also nice to see that the authors' implementation is efficient and doesn't produce too much computation overhead to the gradient computation under reasonable settings ($C=10$).
> >
> > As for the generalization beyond top-$C$ eigenspace, I understand that top-$C$ eigenspace is empirically shown to be significant, e.g., gradient descent mostly happens in this subspace. Since the running time of ViViT doesn't increase much for larger $k$, the result of this paper would be stronger if it could explain or explore more phenomena beyond the top-$C$ eigenspace. For instance, there are works showing that the neural network Hessian eigenspectrum has more fine-grained structures[1].
> >
> > After reading the reviews from Reviewer bWHi and Wsxx, I agree with them that some applications of ViViT mentioned in this paper seem speculative. Providing convincing and concrete use cases of ViViT would definitely make the result stronger.
> >
> > [1] Papyan, Vardan. "Measurements of three-level hierarchical structure in the outliers in the spectrum of deepnet hessians." arXiv preprint arXiv:1901.08244 (2019).

---

### Review · Reviewer_bWHi · 2022-11-12

**Summary Of Contributions:**

The authors propose ViViT: a method for calculating the generalized Gauss-Newton (GGN) approximation of the Hessian.
Their method is exact, computationally efficient, and gives access to first and second-order directional derivatives of single samples.

**Audience:**

Yes

**Broader Impact Concerns:**

--

**Claims And Evidence:**

Yes

**Requested Changes:**

Suggestions for rephrasing:
* Page 1: *There is evidence that this has led to stagnation in the performance of the first-order optimizers.* Can you elaborate on the causes of stagnation? How is this also relevant for second-order optimizers?
* Page 2: *Finally, we use the previously inaccessible properties of the GGN to study how it is affected by noise during training.* As the evolution of the GGN during training is not central to the paper, --during training-- here adds another layer of noise that is not addressed.
* Page 6: *These insights substantiate ViViT's value as a monitoring tool* Insights are not rigorous to validate the proposed method, this has to be shown through concrete examples.

**Strengths And Weaknesses:**

Strengths:
* well-written paper
* time complexity and wall time are better than the power iteration method in the case of mini-batches

Weaknesses:
* while the paper allows access to higher-order derivatives on a single sample basis, it does not specify any use cases apart from broad descriptions (monitoring tool, second-order optimization, etc.). The lack of concrete examples showing the benefit of this approach is a major weakness as.

---

> ### Author Response · Authors · 2022-11-17
> **Rebuttal**
>
> Dear Reviewer bWHi,
>
> Thank you for your suggestions and feedback, some of which we have already
> integrated into our updated manuscript.
>
> We understand your criticism that we only touch on possible applications for the
> quantities we make available in this work. For instance, training neural
> networks with our curvature is one desirable direction for future work, but not
> the immediate goal of our paper. Instead, we tackle the necessary intermediate
> step of making the required information—one, but not the only ingredient in a
> second-order method—available through an efficient implementation. We
> demonstrate its performance and focus on investigating the setting in which such
> an optimizer operates through the newly accessible quantities, like curvature
> noise.
>
> Please find our comments on the other points you raised below:
>
> - *"Can you elaborate on the causes of stagnation?"*
>
>     Of course, it is hard to pin down the exact cause why newly proposed
>     first-order methods have not led to significant improvement over existing
>     methods over the last years [1]. We believe that one aspect is that existing
>     and new methods are functionally very similar in that they heavily rely on
>     the average mini-batch gradient, because it can be computed efficiently.
>
>     Second-order methods are state-of-the-art in 'classical' optimization and
>     functionally different from gradient-based methods. Therefore, they might be
>     able to overcome the stagnation of currently popular first-order methods.
>     However, second-order methods are not as explored as first-order methods in
>     the context of deep learning. To investigate second-order methods, we need novel
>     ways to efficiently compute with curvature. Our work tackles this and is
>     insofar a step forward to explore the potential of second-order methods for
>     deep learning.
>
>     Please let us know in case you have follow-up questions about this
>     clarification.
>
> - *"--during training-- here adds another layer of noise that is not addressed"*
>
>     We would like to address your concern but are unsure how to interpret it.
>     Could you clarify your statement? Specifically, what do you mean by "noise
>     that is not addressed"?
>
> - *"Insights are not rigorous to validate the proposed method, this has to be
>     shown through concrete examples"*
>
>     As outlined above, we understand this criticism, and also believe that the
>     applications we name in the main text will be important directions for
>     future work. However, we believe that they are beyond the scope of this
>     work, which is concerned with the computational approach to efficiently
>     extract quantities that provide the foundation for such paths. As a concrete
>     example, we studied the newly available quantities during training. We are
>     not aware of other works that have previously investigated these quantities
>     we study in this paper due to their prohibitively expensive computation.
>
>     We have adapted the phrase "These insights substantiate ViViT's value
>     as a monitoring tool" to  "[...] where it can serve as a monitoring tool of
>     novel quantities that have not been explored previously to analyse training
>     and other phenomena."
>
> We would be happy to further discuss.
>
> ---
>
> [1] Schmidt, R. M., Schneider, F., & Hennig, P. (2021). Descending through a
>   crowded valley - benchmarking deep learning optimizers.

---

> ### Comment · Reviewer_bWHi · 2022-12-19
> **update after author response**
>
> I thank the authors for their detailed response.

---

### Review · Reviewer_Wsxx · 2022-11-13

**Summary Of Contributions:**

This paper describes an efficient way to compute the eigenvalue decomposition of the minibatch Gauss-Newton matrix -- information which could potentially be useful for scientific or algorithmic (second order optimization) purposes.  The Gauss-Newton matrix is a positive semidefinite approximation to the Hessian which consists of (a) gradients of the network, and (b) second derivatives of the loss function.  One might be interested in computing the eigenvectors and eigenvalues of this matrix over a minibatch.

One way to compute the top eigenvalues and eigenvectors of the minibatch GNVP would be to run the power method with an oracle that computes matrix-vector products between the Gauss-Newton matrix and arbitrary vectors (easy to write in PyTorch).  In contrast, this paper presents an _alternative_ method which (a) can deliver _all_ of the eigenvectors of the minibatch GN matrix, rather than just the top ones, and (b) is apparently faster at even computing just the top ones.

The basic idea is to leverage the fact that the minibatch Gauss-Newton matrix, which has dimension $D \times D$, where $D$ is the number of network parameters, is related to a Gram matrix of dimension $NC \times NC$, where $N$ is the minibatch size and $C$ is the number of classes.  (If the loss function is mean squared error, this Gram matrix is the neural tangent kernel.) In particular, the GN matrix is $\mathbf{V} \mathbf{V}^T$ and the Gram matrix is $\mathbf{V}^T \mathbf{V}$, for a certain matrix $\mathbf{V}$ of shape $D \times NC$.  The GN matrix and the Gram matrix share non-zero eigenvalues, and given an eigenvector of one of these matrices, we can recover the corresponding eigenvector of the other matrix by multiplying by $\mathbf{V}$ or $\mathbf{V}^T$.  Since $NC \ll D$ in settings of interest, if we want to compute the eigenvectors of the GN matrix $\mathbf{V} \mathbf{V}^T$, it is computationally more efficient to compute the eigenvectors of the Gram matrix $\mathbf{V}^T \mathbf{V}$, and then multiply by $\mathbf{V}$ to get the eigenvectors of the GN matrix.

If one were using PyTorch plain's autograd engine, computing  $\mathbf{V}$  would cost $NC $ times the cost of the minibatch gradient.  But helpfully, by leveraging the "Backpack" pytorch library (which provides access to the gradients for individual examples in the minibatch), the matrix $\mathbf{V}$ can be computed using only $C$ times the cost of computing the minibatch gradient.  (So, if the network had just one output unit, then $\mathbf{V}$ could be computed at no additional cost over computing the minibatch gradient; for $C > 1$, the authors propose an approximation that retains this time complexity.)

To compute the eigenvalue spectrum of the minibatch Gauss-Newton matrix, the authors therefore propose: (a) compute the minibatch $\mathbf{V}$ using Backpack, at $C$ times the cost of computing the minibatch gradient; form the $NC \times NC$ Gram matrix $\mathbf{V}^T \mathbf{V}$; compute the eigendecomposition of this matrix; recover the eigenvectors of the Gauss-Newton matrix by multiplying the Gram matrix eigenvectors by $\mathbf{V}$.

The authors show that this method (called "ViViT") is faster in wall clock time than the baseline power method approach that I described above, they discuss and test several approximations to speed up ViViT further, and they study the extent to which the top eigenspace of the minibatch GN matrix is similar to that of the population GN matrix.


**Audience:**

Yes

**Claims And Evidence:**

Yes

**Requested Changes:**

- I found it confusing that the letter $N$ was used to denote both the size of the training dataset and the size of the minibatch.  The paper should be explicit about when $N$ is intended to denote the training dataset size, and when $N$ is intended to denote the minibatch size.  Notably, Section 2 seems to use $N$ to mean the size of the training dataset, yet Footnote 1 in Section 2 asserts that modern NNs are always in the 'overparameterized' setting, which is only true if $N$ means the minibatch size.  (e.g. the ResNet-32 studied in this work has 464k parameters, which is 'underparameterized' since the dataset size is 50k and the num classes is 10.)

- In Appendix B.1, what does the column called "mc, full" mean?  I don't believe that this label was explained.

Optional, might strengthen the work:
-  In section 3.1, when the authors assess the wall clock performance of ViViT on the task of computing the top eigenpairs of the minibatch Gauss-Newton matrix, it might be interesting to also test the approach of (1) explicitly computing the gram matrix $\tilde{\mathbf{G}}$ and (2) running power method on that gram matrix.  You could then compare that approach to the baseline (already in the paper) of running power method directly on $\mathbf{G}$ using GNVP products; these approaches are directly comparable since both are based on the power method, and hence, for a fair comparison, you just need to make sure that the power method tolerance is set to the same value for both methods.  By contrast, in the current draft, making a comparison between ViViT and the baseline is a bit of a headache because by changing the power method tolerance, you can speed up or slow down the baseline arbitrarily (as noted on p. 17).  This is just a suggestion.

**Strengths And Weaknesses:**

Strengths
- I think it's far from obvious that this method of computing the top eigenvalues and eignvectors of the minibatch GN would be much faster, in wall clock time, than running power method using a GNVP oracle, as the authors show in Figure 2.  Leveraging Backpack seems to be crucial here for the performance.
- It's nice to see an examination of several approximations which further bring down the time complexity of ViViT.
- It's nice to see a study comparing the top eigenspace of the minibatch GGN to that of the population GGN (Figure 3b).  Ap

Weaknesses
- The method cannot scale to large minibatch sizes, and it's unclear whether the Gauss-Newton spectrum computed over small minibatches is useful, since this matrix can differ substantially from the population version (e.g. see Figure 3(b)).
- Any applications to second-order optimization seem quite speculative.
- The authors say that a benefit of ViViT is that it provides access to the "example-wise" directional curvatures.  That is, for some direction $\mathbf{e}$, by using the baseline approach of Gauss-Newton vector product on the minibatch we could compute the overall minibatch directional curvature $\mathbf{e}^T \mathbf{G} \mathbf{e}$, but by using ViViT we could get not only that but also the "example-wise" curvatures $\mathbf{e}^T \mathbf{G}_i \mathbf{e}$ for each example $i$ in the minibatch, where $\mathbf{G} = \sum_i \mathbf{G}_i$.  (I think this is also what the authors are getting at in the abstract when they write that "existing works ... do not consider noise in the mini-batch."). However, it's not clear to me what is the benefit of knowing these example-wise curvatures.
- It's not clear to me that the "MC approximation" approach of retaining $M$ (out of $C$) classes from each datapoint is better than just naively subsampling an equivalent number of terms from the $\sum_n \sum_c$ double sum.  Could this authors comment on this?

---

> ### Author Response · Authors · 2022-11-17
> **Rebuttal**
>
> Dear reviewer Wsxx,
>
> Thank you very much for your thoughtful and detailed feedback.
>
> **Requested changes:**
> - The column label "mc, full" in Figures S.4 and S.5 corresponds to "mb, mc" in
> Table S.1: It refers to the MC-approximated GGN with all mini-batch samples. We
> also incorporated your suggestion of distinguishing the training set from the
> mini-batch size in our notation. We have uploaded an updated version of our
> manuscript in which both shortcomings are resolved.
> - Thanks a lot for your interesting suggestion of applying the power iteration
> to both the Gram matrix and the GGN. We agree that this comparison would be
> fairer and easier to interpret. We are working on realizing this comparison and
> will soon post another comment with the results.
>
> **Discussion:**
> - *Limitations:* You’re right that our method, in its exact form, is limited to
> cases where $NC$ is "small". However, sub-sampling (small $N$) is an
> indispensable technique in stochastic optimization used by all practical first-
> and second-order methods. Hence, whether the batch sizes that ViViT can handle
> are considered "large enough" will depend on the application. For training, we
> show in our work that the critical batch sizes exceed the traditionally used
> batch sizes. Additionally, the memory overhead can be further reduced
> significantly using ViViT’s approximations.
>
>     Examples, where small mini-batches yield useful curvature information, are subsampled Hessian-free Newton methods: these methods use particularly small mini-batches for the curvature estimate [1, Section 6.1.1]. Figures S.24 and S.25 provide another hint that curvature might be useful, even on mini-batches: the curvature signal is consistently stronger than the gradient signal.
>
> - *Second-order optimization:* We agree that training neural networks with our
> curvature model is one desirable direction for future work. Our paper takes one
> necessary intermediate step towards this goal: the information we make
> efficiently accessible enables the community to better understand the setting in
> which such an optimizer operates. And empirical analyses like ours also allow
> for identifying challenges like curvature noise.
>
>     The per-sample quantities represent the distributions of slopes and curvatures and are therefore a much richer object than their expected value: they tell us about the reliability of that information. We believe that these distributions can eventually be used for e.g. variance-adapted second-order optimization and thus could lead to more robust and efficient algorithms.
>
> - *Approximations:* You are right: In principle, one can use an arbitrary subset
> of terms from the double-sum over $n$ and $c$ to obtain an estimate of the GGN.
> The sampling techniques we present are well-established in the literature:
> curvature sub-sampling is a common technique in both theoretical [2] and
> practical [3] works on second-order methods. For the loss functions studied in
> our work, the MC approximation of the GGN corresponds to a principled scheme for
> approximating the Fisher information matrix by sampling from the model's
> likelihood implied by the loss. This scheme is used by seminal works on
> approximating the Fisher/GGN [4, 5]. The MC-approximated quantities have been
> found to work well empirically in the context of second-order optimization [6].
>
> Please let us know if we can clarify things further.
>
> ---
>
> [1] Bottou, L., Curtis, F. E., & Nocedal, J. (2016). Optimization methods for
>   large-scale machine learning.
>
> [2] Roosta-Khorasani, F., & Mahoney, M. W. (2019). Sub-sampled newton methods.
>
> [3] Zhang, H., Xiong, C., Bradbury, J., & Socher, R. (2017). Block-diagonal
>   Hessian-free optimization for training neural networks.
>
> [4] Martens, J., & Grosse, R. (2015). Optimizing neural networks with
>   Kronecker-factored approximate curvature.
>
> [5] Grosse, R., & Martens, J. (2016). A kronecker-factored approximate Fisher
>   matrix for convolution layers.
>
> [6] Dangel, F., Kunstner, F., & Hennig, P. (2020). BackPACK: packing more into
>   backprop.

---

> > ### Author Response · Authors · 2022-11-21
> > **Additional results: Power iteration on the Gram matrix**
> >
> > Dear Reviewer Wsxx,
> >
> > we followed your suggestion and performed another run time comparison for
> > computing the GGN's leading $k$ eigenpairs. Instead of computing the full
> > spectrum of the GGN Gram matrix, we used the same power iteration as the
> > baseline (power iteration with GGN-vector products).
> >
> > We have updated the manuscript, which now includes a new figure (S.7) and a
> > paragraph *"Power iteration on the GGN versus power iteration on the GGN Gram
> > matrix"*. The figure presents results for the same setting as the run time
> > comparison in the main text (Figure 2b).
> >
> > In general, this additional experiment confirms the findings of the comparison
> > in the main text: ViViT, even without approximations, outperforms the
> > power iteration for $k \ge 2$.
> >
> > Additionally, we found that when the Gram matrix has sufficiently large
> > dimension, the power iteration requires less matrix-vector products than the
> > eigensolver that computes the full spectrum and therefore further improves the
> > run time of our approach. However, if the Gram matrix is small, the simplistic
> > power iteration requires more matrix-vector products than a sophisticated
> > eigensolver and leads to longer run times.
> >
> > Please let us know if you have feedback on this additional experiment.

---

### Decision · Action_Editors · 2023-01-04

**Recommendation:** Accept as is

**Comment:**

This manuscript is clearly written, addresses a topic of wide interest in the machine learning community (second order information in large-scale optimization), and delivers interesting (empirical) computational contributions.  Among the weaknesses raised by reviewers are lack of theoretical guarantees and of experimental evaluations in large-scale settings.  While I agree with these critiques, I do not think they deem the manuscript unsuitable for publication, particularly since the text is transparent about said limitations.  A more significant shortcoming, raised by multiple reviewers, is that the paper does not provide sufficient use cases for the quantities it allows computing.  The authors provide certain (rather speculative) possibilities for use cases, which I believe suffice for publication.  However, the work will be much stronger if it were to include more concrete suggestions.  I encourage the authors to address this point in the camera-ready version of the text.

**Audience:**

Yes, second order information in large scale optimization is of interest to the machine learning community.  Methods that allow extracting such information efficiently are definitely in need.

**Claims And Evidence:**

Yes, the paper is transparent about the nature of its contributions, and those are supported properly.